# Early life imprints the hierarchy of T cell clone sizes

**Mario U Gaimann[1,2], Maximilian Nguyen[1], Jonathan Desponds[3], Andreas Mayer[1]\***

[1]Lewis-Sigler Institute for Integrative Genomics, Princeton University, Princeton, United States; [2]Arnold Sommerfeld Center for Theoretical Physics and Center for NanoScience, Department of Physics, Ludwig-Maximilians-Universität München, München, Germany; [3]NSF-Simons Center for Quantitative Biology, Northwestern University, Evanston, United States

**Abstract** The adaptive immune system responds to pathogens by selecting clones of cells with specific receptors. While clonal selection in response to particular antigens has been studied in detail, it is unknown how a lifetime of exposures to many antigens collectively shape the immune repertoire. Here, using mathematical modeling and statistical analyses of T cell receptor sequencing data, we develop a quantitative theory of human T cell dynamics compatible with the statistical laws of repertoire organization. We find that clonal expansions during a perinatal time window leave a long-lasting imprint on the human T cell repertoire, which is only slowly reshaped by fluctuating clonal selection during adult life. Our work provides a mechanism for how early clonal dynamics imprint the hierarchy of T cell clone sizes with implications for pathogen defense and autoimmunity.

## Introduction

The hallmark of adaptive immunity is the generation of diversity through genetic recombination and clonal selection. Their interplay balances the breadth and specificity of the $\sim 10^{12}$ T cells in the human body (*Figure 1A*; *Arstila et al., 1999*; *Farber et al., 2014*): The genetic recombination of the T cell receptor (TCR) locus, termed VDJ recombination, generates an enormous potential diversity of receptors ranging from early estimates of $\sim 10^{15}$ (*Davis and Bjorkman, 1988*) to more recent estimates of $\sim 10^{61}$ (*Mora and Walczak, 2016*) different possible receptor TCRαβ heterodimers. Clonal selection expands the number of specific cells during an infection for effector functions, a fraction of which is retained over prolonged periods of time as immune memory (*Ahmed and Gray, 1996*; *Farber et al., 2014*).

Much progress has been made deciphering the mechanisms of regulation and control of T cell dynamics over the last few decades (*Antia et al., 2005*; *Sallusto et al., 2010*; *Farber et al., 2014*). However, much of that progress has focused on the dynamics of subsets of T cells specific to a particular antigen and has come from experiments in mice. An important open question is how exposures to many antigens over a human lifetime collectively shape our T cell repertoire (*Farber et al., 2014*; *Davis and Brodin, 2018*).

High-throughput repertoire sequencing enables direct surveys of the diversity and clonal composition of T cells from human blood or tissue samples and thus promises to provide quantitative answers to this question (*Robins et al., 2009*; *Thomas et al., 2014*; *Britanova et al., 2016*; *Emerson et al., 2017*; *Oakes et al., 2017*; *Thome et al., 2016*; *Robins et al., 2010*; *Qi et al., 2014*; *Lindau et al., 2019*; *TRACERx consortium et al., 2019*). However, while the TCR locus provides a natural barcode for clonal lineages due to its large diversity, this same diversity also makes inferring past clonal dynamics a challenging inverse problem, in particular given practical limitations on sequencing depth and temporal resolution in longitudinal studies. Mathematical modeling can help

**\*For correspondence:** andimscience@gmail.com

**Competing interests:** The authors declare that no competing interests exist.

**eLife digest** The human immune system develops a memory of pathogens that it encounters over its lifetime, allowing it to respond quickly to future infections. It does this partly through T cells, white blood cells that can recognize different pathogens. During an infection, the T cells that recognize the specific pathogen attacking the body will divide until a large number of clones of these T cells is available to help in the fight. After the infection clears, the immune system 'keeps' some of these cells so it can recognize the pathogen in the future, and respond quicker to an infection.

Over the course of their lives, people will be infected by many different pathogens, leading to a wide variety of T cells that each respond to one of these pathogens. However, it is not well understood how various infections throughout the human lifespan shape the overall population of different T cells.

Gaimann et al. used mathematical modelling to study how the composition of the immune system changes in people of different ages. Different populations of T cells – each specialized against a specific antigen – had been previously identified through genetic sequencing. Gaimann et al. analyzed their dynamics to show that many of the largest populations originate around birth, during the formation of the immune system.

These findings suggest a potential mechanism for how exposure to pathogens in infancy can influence the immune system much later in life. The results may also explain variations in how people respond to infections and in their risk of developing autoimmune conditions. This understanding could help develop new treatments or interventions to guide the immune system as it develops.

address this challenge by solving the forward problem of linking clonal dynamics to emergent statistical patterns (*Desponds et al., 2016*; *Lythe et al., 2016*; *Dessalles et al., 2019*; *Altan-Bonnet et al., 2020*; *de Greef et al., 2020*). Comparing patterns to data can provide insights about dynamics from static snapshots of repertoire organization in different individuals. A particularly striking such pattern has been the observation of power-law scaling of clone sizes spanning several orders of magnitude (*Robins et al., 2009*; *Thomas et al., 2014*; *Britanova et al., 2016*; *Emerson et al., 2017*; *Oakes et al., 2017*). In a typical sample of T cells from peripheral blood, a large fraction of clones is only seen once within $10^5$–$10^7$ sampled sequences, while the most abundant clones account for more than 1% of all sequencing reads. Such power-law scaling of clone sizes has been shown to arise at steady state in models of fluctuating clonal selection driven by different antigen encounters (*Desponds et al., 2016*). However, it is unclear whether this mechanism alone is sufficient to explain how clone size scaling is established, and more broadly how variable the clonal hierarchy is over time.

Here, we develop a theory of T cell dynamics throughout the human lifespan based on the statistical laws of repertoire organization and dynamics revealed by cohort and longitudinal human TCR repertoire sequencing (*Britanova et al., 2016*; *Emerson et al., 2017*; *Chu et al., 2019*; *Lindau et al., 2019*). We find that clonal expansions during repertoire formation play an important role in establishing the clone size hierarchy, which is only slowly reshaped by fluctuating clonal selection during adult life.

## Results

### A scaling law of human T cell repertoire organization

An important statistic to summarize repertoire organization is the clone size distribution, which tabulates the number of clones found at different multiplicities within a repertoire or sample. Multiple previous studies have shown that these distributions are heavy-tailed (*Robins et al., 2009*; *Thomas et al., 2014*; *Britanova et al., 2016*; *Emerson et al., 2017*; *Oakes et al., 2017*). However, potential confounding by noise introduced during the sequencing process has remain debated (*Altan-Bonnet et al., 2020*) and systematic analyses of how variable these distributions are across healthy individuals have been lacking. To fill these gaps, we reanalyzed data from two large-scale cohort repertoire sequencing studies of human blood samples, which used fundamentally different

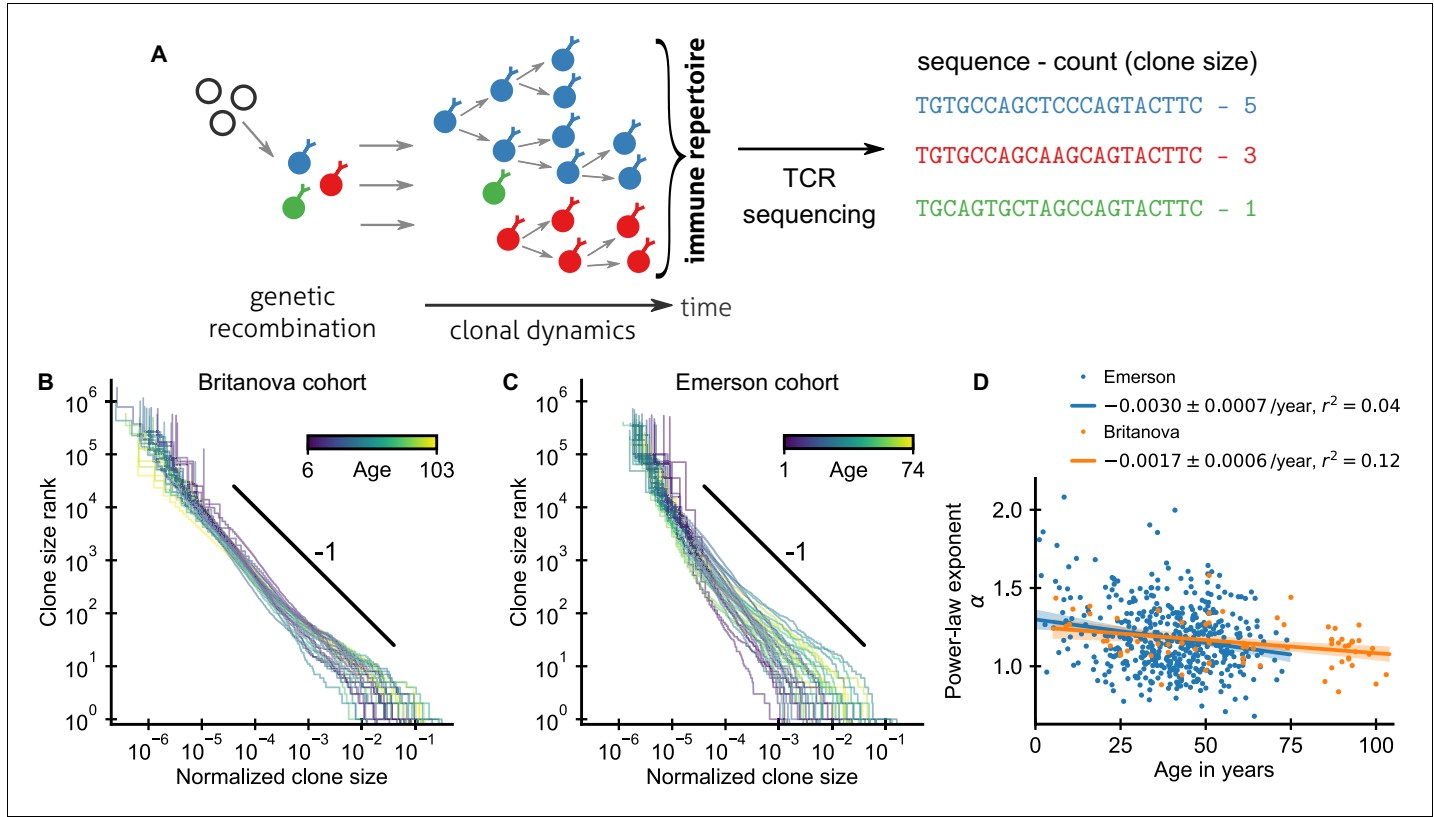

**Figure 1.** Statistics of human T cell repertoire organization. (**A**) T cells with highly diverse receptors are created from progenitor cells through genetic recombination (left), which then undergo clonal selection (middle) together shaping the immune repertoire. The T cell receptor (TCR) locus acts as a natural barcode for clonal lineages, which can be read out by sequencing (right). (**B, C**) Clone size distributions in two large cohort studies of human blood samples using disparate sequencing protocols display a power-law relationship between the rank and size of the largest clones. Each line shows the size distribution of all T cell clones in an individual in an unsorted blood sample, that is independently of the phenotypes of the cells making up the different clones. Ages are color coded as indicated in the legend. The black line shows a power law with a slope of -1 for visual comparison. Normalized clone sizes were defined as the number of reads of a given receptor's sequence divided by the total number of reads within a sample and a factor equal to the average fraction of T cells with memory phenotype at different ages to account for variations in sampling depth and in the subset composition of peripheral blood, respectively (*Figure 1—figure supplement 3*). Only a single individual is displayed per 2-year age bracket to improve visibility. (**D**) Power-law exponents as a function of the age (legend: linear regression slope and coefficient of determination). Data sources: *Britanova et al., 2016*, *Emerson et al., 2017*.

The online version of this article includes the following figure supplement(s) for figure 1:

**Figure supplement 1.** Cohort age distributions.
**Figure supplement 2.** Clone size distributions in phenotypically sorted T cell subsets.
**Figure supplement 3.** Influence of normalization procedure on clone size distributions.
**Figure supplement 4.** Dependence of power-law exponent on age by cytomegalovirus (CMV) infection status and sex.
**Figure supplement 5.** Clone size distributions in cordblood.

sequencing pipelines and thus have different sources of noise (Materials and methods – Experimental data sources). Both studies sequenced the locus coding for the hypervariable TCR CDR3 β-chain from peripheral blood T cells of healthy human volunteers spanning a large range of ages (*Figure 1—figure supplement 1*). In both studies, T cells were sequenced without regard to their phenotypic characteristics. A clone thus represents the full lineage of cells derived from a common ancestral cell irrespective of differentiation status (see Appendix 5 – Relation between clone size and cellular phenotypes and *Figure 1—figure supplement 2* for a phenotypically resolved analysis).

After normalizing clone sizes to account for variations in sampling depth and for the increasing fraction of T cells of memory phenotype with age (*Figure 1—figure supplement 3*), we found that the tails of the clone size distributions collapsed to the same statistical law across individuals and

cohorts (*Figure 1B,C*): Ranking clones by decreasing size, the rank of the largest clones approximately scales with their size $C$ as a power law,

$$\text{rank} \sim C^{-\alpha}, \tag{1}$$

where $\alpha$ is a scaling exponent. To quantify the apparent similarity of the scaling relationship, we determined $\alpha$ for each sample by maximum likelihood estimation. Only a small fraction of all T cells are sampled, which poses a challenge because subsampling a power law leads to deviations from scaling at small clone sizes (*Stumpf et al., 2005*). To overcome this challenge, we used a trimming procedure and excluded clones smaller than a minimal size from the fitting, which decreases bias arising from subsampling (Appendix 1). Determined in this subsampling-robust manner the fitted power-law exponents agree remarkably well within the range of ages covered by both cohorts (*Figure 1D*): with $\alpha = 1.17 \pm 0.03$ (mean ± standard error [SE]) and $\alpha = 1.18 \pm 0.01$ in the Britanova cohort and Emerson cohort, respectively. Moreover, the fitted exponents varied little between individuals in both cohorts; with a sample standard deviation of fitted exponents of 0.14 and 0.21, respectively. The agreement of the mean exponents is noteworthy given the different sequencing pipelines and provides strong evidence that the scaling relationship (*Equation 1*) is a true feature of the clone size distribution and not of the measurement process.

What drives the emergence of a power-law distributed hierarchy of clone sizes? Given the reproducibility of the scaling law across individuals, we might hope for a statistical explanation independent of the precise antigenic history that has driven the expansion of specific cells in an individual. To test hypotheses about mechanisms underlying scaling, we describe repertoire dynamics using a general mathematical framework based on effective stochastic rate equations for the recruitment of new clones, and the proliferation and death of already existing clones within a T cell compartment (Materials and methods – Mathematical framework). In macroecology, where such reductionist approaches have a long history, simple neutral models within this framework have had surprising success in describing species abundance distributions only accounting for demographic stochasticity

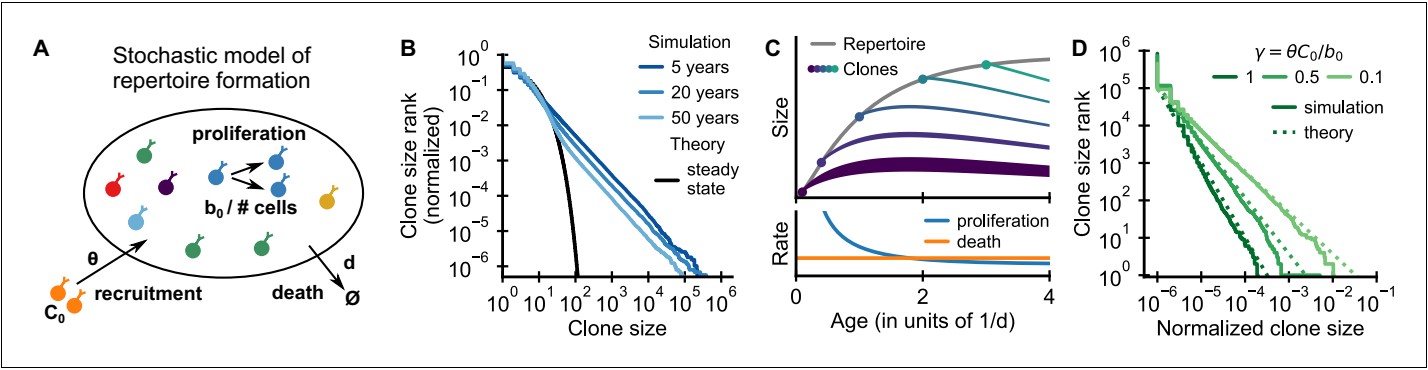

**Figure 2.** Emergence of power-law scaling of clone sizes in a minimal model of repertoire formation. (**A**) Sketch of the stochastic dynamics of recruitment, proliferation, and death of T cells. Proliferation is inversely proportional to total repertoire size, which reflects increased competition as the repertoire grows. (**B**) Clone size distributions in simulated repertoires display power-law scaling (blue lines), in contrast to steady-state predictions that conform with those of a null model based only on demographic stochasticity (black line, *Equation 48*). (**C**) Illustration of the mechanism: early in life rates of proliferation exceed clonal turnover (lower panel). As the total repertoire size increases (gray line, upper panel) the proliferation rate decreases due to increased competition. The dynamics of selected clones after their recruitment marked by a dot is indicated by colored lines (upper panel). The line position shows the cumulative size of all prior clones, while the line width indicates the size of the clone (not to scale). The earlier a clones is recruited the larger it expands during the period of overall repertoire growth. (**D**) Dependence of the clone size distribution on parameters. Simulated repertoires at 5 years of age were subsampled to $10^6$ cells to mimic the experimental sampling depth (solid lines). The simulated data closely follow predictions from a continuum theory of repertoire formation (dashed lines). Model parameters: (B,D) clonal death rate $d = 0.2$/year, clonal recruitment rate $\theta = 10^6$/year, clone size at recruitment $C_0 = 1$; (B) total proliferation rate $b_0 = 10^7$/year (implying a recruitment to proliferation ratio $\gamma = 0.1$), (D) variable $b_0$ as indicated in the legend by the ratio $\gamma$.

The online version of this article includes the following figure supplement(s) for figure 2:

**Figure supplement 1.** Analytical predictions for clone size distributions in a model with variable recruitment sizes.
**Figure supplement 2.** Simulated clone size distributions in a model with saturation of proliferation rates.
**Figure supplement 3.** Simulated clone size distributions in a model with competition for clone-specific resources.

(*Volkov et al., 2003*), but this source of variability is insufficient to account for the observed breadth of T cell clone sizes (*Desponds et al., 2016*; *de Greef et al., 2020*; for a detailed discussion see Appendix 2). The failure of this null model has prompted a search for other mechanisms that explain scaling.

To constrain this search, we analyzed how fitted exponents varied with age. In particular, based on a finite time solution we derived for a previously proposed model (*Desponds et al., 2016*) of how power-law scaling can emerge from the cumulative effect of temporal fluctuations in clonal growth rates (Materials and methods – Slow convergence to steady–state scaling), we expected a substantially steeper tail in young individuals. While exponents overall decreased slightly with age, the dependence on age accounted for surprisingly little variation in both cohorts (*Figure 1D*), including when controlling for sex and cytomegalovirus (CMV) exposure status (*Figure 1—figure supplement 4*). Notably, scaling is established within the first decade of life, with significant clone size variability existing as early as at birth (*Figure 1—figure supplement 5*), defying previous model predictions.

## A mechanism for the emergence of scaling during repertoire formation

We hypothesized that scaling might result from clonal expansions during repertoire formation, which would naturally explain the early onset of scaling. Our hypothesis is based on experimental evidence in mice (*Le Campion et al., 2002*; *Min et al., 2003*; *Haluszczak et al., 2009*; *Kawabe et al., 2017*) and human (*Rufer et al., 1999*; *Schönland et al., 2003*) that repertoire formation is driven not only by increased thymic output, but also by large proliferative expansion of some T cell clones. Additionally, multiple studies *Hammarlund et al., 2003*; *Pogorelyy et al., 2017*; *Tanno et al., 2020* have shown that some T cell clones can persist over multiple decades, which suggests that clonal turnover might be sufficiently slow for transient expansionary dynamics early in life to shape repertoire organization over a prolonged time period (see also Appendix 2 –Relaxation time scale in a neutral model).

To test our hypothesis, we constructed a minimal model of repertoire formation based on known T cell biology (*Figure 2A*). Following previous work (*Bains et al., 2009*; *Lythe et al., 2016*), we assume that T cells proliferate at a rate inversely proportional to the total number of cells $N$ already present in the repertoire. This assumption leads to increased proliferation early in life before the repertoire has reached its homeostatic size, for which there is experimental evidence *Le Campion et al., 2002*; *Min et al., 2003*; *Haluszczak et al., 2009*; *Kawabe et al., 2017*; *Rufer et al., 1999*; *Schönland et al., 2003*. This assumption is also compatible with a simple mechanistic model of T cell competition (Materials and methods – Mechanistic motivation for the competition function). We further assume that cells die at a rate $d$, and that new clones are recruited at rate θ with an initial size $C_0$. For simplicity, we set $C_0$ equal to one in the following and we assume constant rates $d$ and θ. Importantly, recruitment of new clones and total expansion of already existing clones maintain a constant ratio throughout development under these assumptions in line with findings that the fraction of cells with TCR excision circles, which are diluted during peripheral division, is constant during fetal development (*Schönland et al., 2003*) and infancy (*Douek et al., 2001*; *Bains et al., 2009*).

We simulated the model starting from an empty repertoire and found that large clones displayed power-law scaling (*Figure 2B* blue lines). The simulation results contrast with steady-state predictions (*Figure 2B* black line), where the model effectively reduces to the neutral null model introduced earlier (Materials and methods – Steady-state distribution). The power-law tail persisted over multiple decades of aging, much beyond the timescale of cellular turnover, $1/d$ = 5 years, assumed in the simulations. A mathematical analysis shows that relative timescales of clonal and cellular turnover are controlled by the control parameter $\gamma = \theta C_0/b_0$, which is the ratio of the contribution of recruitment and proliferation to overall compartment maintenance (Appendix 2). The long timescale of clonal turnover emerges because the chosen parameters are in the biological parameter regime $\gamma<1$, where most cell death is balanced by proliferation (*den Braber et al., 2012*; *Macallan et al., 2017*). Thus, we find that repertoire formation can produce transient but long-lasting power-law scaling of clone sizes.

To obtain intuitive insight into how scaling is established, we developed a continuum theory of clonal dynamics during repertoire growth (Materials and methods – Continuum theory of clonal growth). We find that the clone size $C_i$ of the $i$-th clone recruited at time $t_i$ follows a subexponential

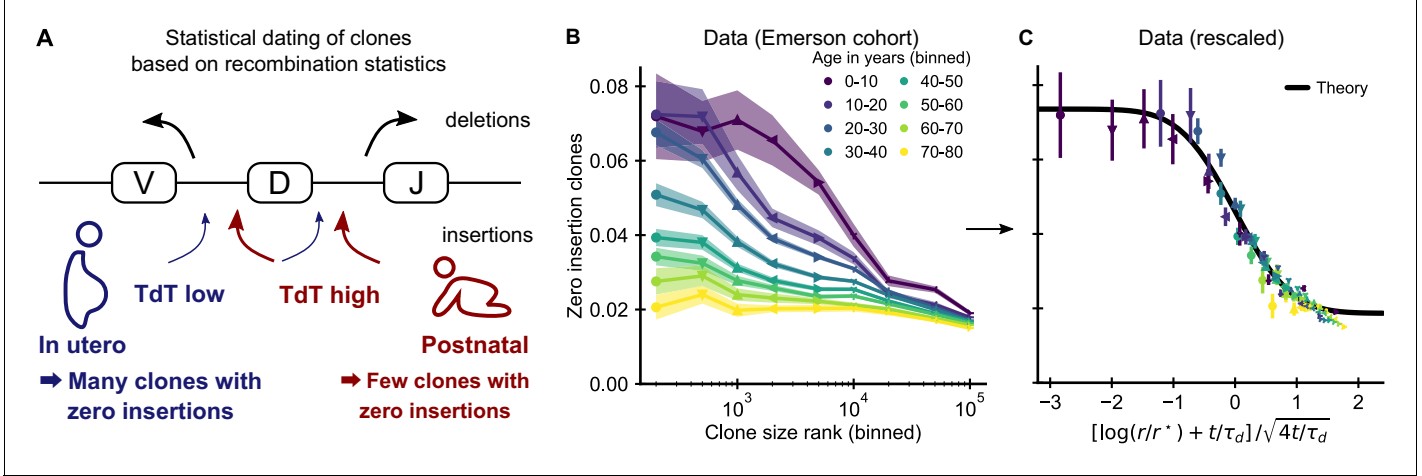

**Figure 3.** Statistical dating of clones reveals that early expansions have a long-lasting effect. (**A**) Genetic recombination of a TCR involves the choice of a V, D, and J region among multiple genomically-encoded templates as well as the deletion and insertion of nucleotides at both the VD and DJ junctions. The enzyme TdT, which is responsible for nucleotide insertions, is not expressed during early fetal development. This allows a statistical dating of clonal ages, as clones with zero insertions at both junctions constitute a much larger fraction of all clones during a fetal and perinatal time window. (**B**) Fraction (± SE) of clones with zero insertions as a function of age and clone size. Clones are binned by their size into non-overlapping bins (rank 1–500, 501–1000, and so on; upper values are indicated on the x-axis). (**C**) Same data as in B displayed with a rescaled x-axis using fitted parameters $\tau_d$ = 9.1 ± 0.5 years, $r^*$ = 1.2 ± 0.2 · $10^4$. The data collapses onto a sigmoidal function predicted by theory (**Equation 3**) with fitted $p_{0,-}$ = 0.074 ± 0.004, $p_{0,+}$ = 0.0187 ± 0.0005 (black line). Data source: **Emerson et al., 2017**.

The online version of this article includes the following figure supplement(s) for figure 3:

**Figure supplement 1.** Fraction of zero insertion clones within productive and unproductive sequences.

**Figure supplement 2.** Enrichment of clones with known specificity.

**Figure supplement 3.** Enrichment of clones with high probability of generation.

**Figure supplement 4.** Fraction of zero insertion clones as function of age and clone size by cytomegalovirus (CMV) infection status.

growth law $C_i(t) = C_0(t/t_i)^{1/(1+\gamma)}$. Clones recruited early grow large deterministically until competition lowers proliferation rates below the death rate (**Figure 2C**, lower panel). Different clones are recruited at different times and thus have more or less time to grow (**Figure 2C**, upper panel), which leads to a clone size distribution that follows power-law scaling with an exponent $\alpha = 1 + \gamma$. We note that this origin of the power-law scaling is closely related to a well-known generative mechanism for power-laws first studied by **Yule, 1924** (for a detailed discussion see Appendix 4 – A unified view on mechanisms generating power laws in different growth processes).

The predicted exponent closely matches simulation results for different values of γ (**Figure 2D** dashed lines). Intuitively, when recruitment rates are higher, clones founded early have less time to outgrow later competitors, and thus the power law is steeper (α is larger). Importantly, in the biological parameter regime in which proliferation dominates, $\gamma<1$, the exponent is compatible with experiments (**Figure 1B–D**). We thus find, that the model – without fine tuning of parameters – reproduces the observed scaling exponent.

To expose a basic mechanism capable of producing broad clone size distributions, we have kept the model deliberately simple. More detailed models demonstrate the conditions and limits on the generalizability of this mechanism (Appendix 3). Variable recruitment sizes only affect the distribution of small clones (**Figure 2—figure supplement 1**); while a saturation of proliferation rates, or competition between subsets of T cells for specific resources maintain distributions at small and intermediate sizes while leading to cutoffs for the largest clones (**Figure 2—figure supplement 2** and **Figure 2—figure supplement 3**, respectively).

## Long-lived incumbency advantage shows early expansions imprint clone size hierarchy

Our proposed theory for the rapid emergence of scaling predicts that large clones have expanded massively during repertoire formation. To test this prediction, we need to trace the dynamics of early

founded clones. To this end, we exploit a change in the recombination statistics taking place during fetal development (*Feeney, 1991*; *Rechavi et al., 2015*; *Park et al., 2020*; *Figure 3A*). While T cells are produced by the thymus from the late first trimester the enzyme terminal deoxynucleotidyl transferase (TdT), which inserts non-templated nucleotides during VDJ recombination, is not expressed until the mid second trimester (*Park et al., 2020*). Therefore, many more T cells in fetal and neonatal blood have zero insertions than expected from the adult recombination statistics (*Rechavi et al., 2015*). This enables a statistical dating of individual clones in a repertoire based on their sequence (*Sethna et al., 2017*; *Pogorelyy et al., 2017*).

If our model is correct, we expect abundant clones to be more likely to have zero insertions than smaller clones. Analyzing data from the Emerson cohort, we find that zero insertion clones are indeed highly enriched within the most abundant clones (*Figure 3B*). This generalizes a previous report of such an enrichment within the naive compartment (*Pogorelyy et al., 2017*). The large cohort size allows us to perform a fine-grained analysis of how the fraction of zero-insertion clones depends on clonal abundance and age. We find that enrichment is particularly pronounced in the young and decreases with age at different speeds depending on clone size. Among the largest clones many more still have zero insertions than expected from the adult recombination statistics even multiple decades after repertoire formation. This suggests that the incumbent large clones created during repertoire formation are only slowly replaced by clones expanding later in life, similarly to what has been observed in mice (*Gossel et al., 2017*; *Hogan et al., 2019*).

Additional analyses rule out other potential explanations for the relation between insertion statistics and clonal abundance. First, sequences with zero insertions are similarly enriched among the largest clones in productive and unproductive sequences (*Figure 3—figure supplement 1*) demonstrating that convergent selection pressures during adult life are not a primary source of the higher abundance of these clones. Second, while abundant clones are also enriched for sequences with known antigen specificity (*Figure 3—figure supplement 2*) and sequences likely to be convergently recombined (*Figure 3—figure supplement 3*), these enrichments do not show the same striking dependence on age. Furthermore, we find that zero insertion clones are consistently less enriched in individuals infected by CMV (*Figure 3—figure supplement 4*), in contrast to the hypothesis that this infection might drive their expansion (*Pogorelyy et al., 2017*). Taken together, these analyses

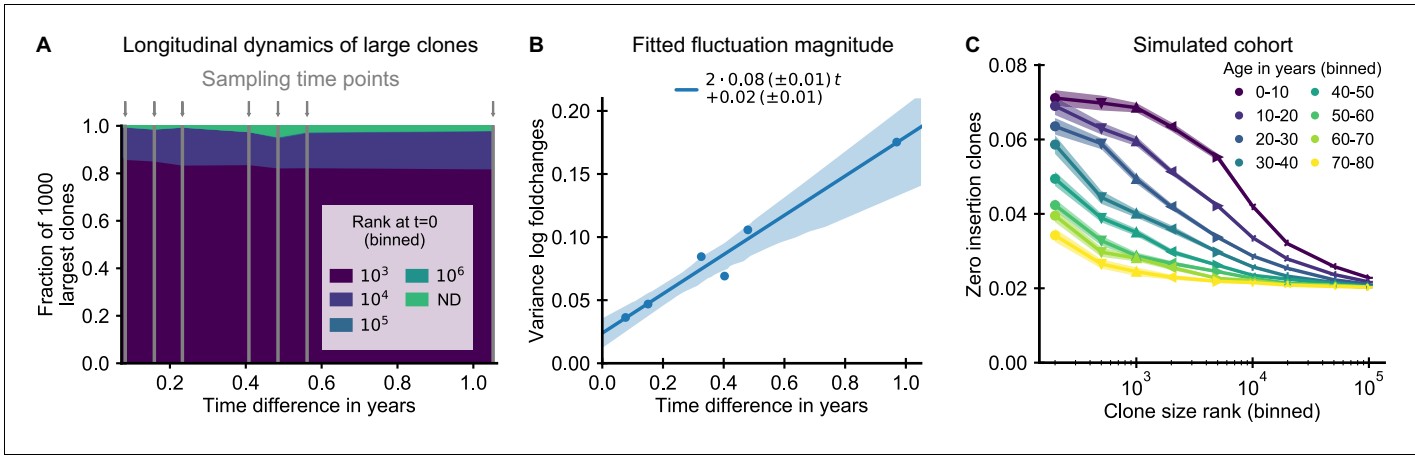

**Figure 4.** The small magnitude of longitudinal clone size fluctuations implies a slow reordering of the clone size hierarchy. (A,B) Longitudinal clonal dynamics in a healthy adult over a one year time span. (A) Fraction of the 1000 largest clones that fall within a specific clone size rank bin at the earliest time point. A small number of clones was not detected at all at the first time point (ND) likely representing recently expanded clones. All other clones were already among the largest clones initially. (B) Variance of log-foldchanges in clone size as a function of time difference for the 250 largest clones. (C) Fraction of clones with zero insertions as a function of age and clone size in a simulated cohort using a magnitude of clonal growth rate fluctuations inferred from the longitudinal data. Data source: *Chu et al., 2019*.
The online version of this article includes the following figure supplement(s) for figure 4:

**Figure supplement 1.** Provenance of large T cell clones for all subjects.
**Figure supplement 2.** Dynamics of large persistent clones for all subjects.
**Figure supplement 3.** Data collapse by parameter rescaling for the simulated cohort.

support the conclusion that dynamics during the perinatal time window of repertoire formation leave a long-lasting imprint on the T cell clonal hierarchy well into adulthood.

## Longitudinal clone size fluctuations predict the dynamics of the clone size hierarchy with aging

Building on this successful validation of a core prediction of our theory, we asked whether we could leverage the detailed pattern of enrichments at different ranks and ages to quantify how much being part of the wave of early expansions determines the fate of a clone relative to other sources of clone size variability. To this end, we extended our model beyond repertoire formation and allowed clonal proliferation rates to fluctuate over time to model the net effect of clonal selection by changing antigenic stimuli during adult life (*Desponds et al., 2016*). Specifically, we added a stochastic term to the growth rate of each clone to provide an effective Langevin description of the long-term dynamics induced by fast-changing antigen stimuli,

$$b_i(t) = b_0/N + \sqrt{2}\sigma \eta_i(t), \tag{2}$$

where $\langle \eta_i(t)\eta_j(t')\rangle = \delta_{ij}\delta(t-t')$.

To determine a biologically plausible fluctuation strength $\sigma$, we analyzed the variability of clone sizes over time in a longitudinal repertoire sequencing study (*Chu et al., 2019*). We first analyzed how much recently expanded clones contribute to the tail of the clone sizes, and found that only a small fraction of the largest clones in any sample were not already large at the earliest time point (*Figure 4A* and *Figure 4—figure supplement 1*). To minimize confounding by transient dynamics affecting these clones, we excluded them from further analysis. We found that large clones had remarkably stable abundances over time, which we quantified by calculating the variance of log-fold-changes in clone size between the second and every subsequent time point (*Figure 4B* and *Figure 4—figure supplement 2*). The variability of clone sizes increased linearly over time as expected theoretically, from which we determined a magnitude of net growth rate fluctuations $\sigma$ compatible with the slope of increase (Materials and methods – Modeling long-term repertoire dynamics with fluctuating clonal growth rates).

Using the fitted fluctuation strength, we constructed an in silico cohort of individuals of different ages according to the extended model (Materials and methods – Simulated cohort). In short, we computationally assigned each newly recruited clone to have zero insertions in a way that mimics the change in fetal recombination statistics, and we simulated memory repertoire dynamics based on the combined effect of early expansion and fluctuating clonal selection (*Equation 2*). The enrichment of zero insertion clones in the simulated cohort (*Figure 4C*) closely recapitulated the empirical findings using plausible parameter values. Notably, the longer lasting enrichment of zero insertion clones among the very largest clones is also found in the simulated cohort, and the timescales over which the enrichment decays agree remarkably well.

To obtain analytical insight into the enrichment dynamics, we studied how fluctuating growth rates reorder the clone size hierarchy established during repertoire formation (Materials and methods – Relaxation of the zero insertion distribution). The early clone size hierarchy in our model is dominated by the time of recruitment, which leads to a steep gradient in the zero insertion probabilities as a function of rank in the first decade of life (*Figure 4C*). We thus calculated how the zero insertion probabilities $P_0(r,t)$ for clones of rank $r$ at time $t$ change due to fluctuating clonal growth starting from an idealized initial condition resembling the early clone size hierarchy, in which the clone sizes follow power-law scaling and the $r^\star$ largest clones have a zero insertion probability $p_{0,-}$ and all others a probability $p_{0,+}$. We find that the probability of zero insertion clones then follows a sigmoidal shape as a function of log clone size rank, which changes with age as follows,

$$P_0(r,t) = \frac{\Delta p_0}{2}\operatorname{erfc}\left(\frac{\log(r/r^\star) + t/\tau_d}{2\sqrt{t/\tau_d}}\right) + p_{0,+}, \tag{3}$$

where $\Delta p_0 = p_{0,-} - p_{0,+}$ is the difference of zero insertion probabilities, $\tau_d$ a characteristic timescale, and $\operatorname{erfc}(x)$ the complementary error function. These analytical results suggest a two-parameter rescaling of the enrichment of zero insertion clones as a general test of our theory. To demonstrate the feasibility of fitting these parameters from the enrichment dynamics, we determined them from

the simulated data using weighted least squares fits setting $r$ and $t$ in *Equation 3* to the mid-value of each bin. Rescaling the data with the fitted $r^*$ and $\tau_d$ lead to a collapse of all simulated datapoints onto the predicted sigmoidal curve (*Figure 4—figure supplement 3*). We then applied the same fitting and rescaling procedure to the experimental data, and found that it also leads to a remarkably good data collapse (*Figure 3C*).

The fitted parameters quantify key features of long-term repertoire dynamics, with $\tau_d$ characterizing the timescale over which fluctuations change the clone size hierarchy, and $r^*$ being related to the number of clones recruited during early repertoire growth. In line with the long-lived enrichment of zero insertion clones, the fitting reveals a remarkably slow timescale of about a decade over which the clone size hierarchy is reordered during healthy aging. The fitted $r^*$ indicates that early repertoire formation involves the expansion of a large number of different clones. Overall, the agreement between theory and data demonstrates that our model quantitatively captures how early expansions and ongoing fluctuating selection together shape the clone size hierarchy.

## Discussion

The evolution of the adaptive immune system has endowed vertebrates with the ability to adapt to pathogens that evolve on a timescale faster than host reproduction (*Mayer et al., 2016*). However, this ability comes with a cost: every generation needs to rebuild immune memory anew. As the organism first comes into contact with the outside world, it quickly needs to train its adaptive immune system to tolerate innocuous antigens and build up immune memory against pathogens. Here, we have shown that this process of rapid adaptation leaves a long-lasting imprint on the organization of the human T cell repertoire. More broadly, we propose a theory of repertoire dynamics that quantitatively describes how early expansions during repertoire formation combine with a lifetime of exposures to cumulatively shape the T cell hierarchy. Notably, we find that the T cell repertoire is remarkably stable over time in adult individuals outside of the punctuated expansions and contractions of specific clones in acute responses. Our study demonstrates that despite its vast complexity, repertoire dynamics is partially predictable by quantitative models. The model predictions can help guide future longitudinal studies, which in turn will allow refinements of modeling assumptions. The current work thus provides a stepping stone toward a detailed quantitative understanding of T cell dynamics that we hope will ultimately power the rational development of immunodiagnostics and therapeutics.

The general mechanism we describe for imprinting in the adaptive immune system provides a unified lens through which to view a number of converging lines of evidence about how a developmental time window shapes adaptive immunity (*Guerau-de-Arellano et al., 2009*; *Farber et al., 2014*; *Gostic et al., 2016*; *Constantinides et al., 2019*; *Li et al., 2019*; *Davenport et al., 2020*; *Hong et al., 2020*). In our model, overall repertoire growth early in life amplifies the effect of any early exposures, as the responding clones continue to proliferate as memory cells and thus are much larger than memory produced from similar exposures after the homeostatic repertoire size is reached. We thus expect early pathogen exposures to be particularly potent, as has been observed in influenza, where disease severity across age cohorts for different strains depends on the first exposure (*Gostic et al., 2016*). Conversely, we expect the presence of tolerizing factors early in life to be particularly crucial during repertoire formation to avoid autoimmunity, as has been observed for the autoimmune regulator gene AIRE, for which expression is only essential during a perinatal time window (*Guerau-de-Arellano et al., 2009*).

A limitation of datasets used in this study is that they do not provide direct information about the phenotypic characteristics of cells belonging to different clones. Repertoire sequencing of phenotypically sorted blood samples shows that the largest clones predominantly consist of cells with memory phenotype (Appendix 5). This indirectly suggests that the clonal expansions during repertoire formation produce memory cells as we have assumed in our simulated cohort (*Figure 4C*). Supporting this interpretation, a substantial number of memory cells circulate in the blood quickly following birth (*Pediatric AIDS Clinical Trials Group et al., 2003*) and recent evidence suggests that memory-like T cells are already generated in particular tissue sites such as the intestine even before birth (*Zhang et al., 2014*; *Li et al., 2019*). However alternatively, early expansions could also set up a broad distribution of naive T cell clone sizes (*de Greef et al., 2020*), whose hierarchy would then need to be roughly maintained during the transition into memory to be compatible with the

observed impact of early expansions on the hierarchy of the most abundant clones. Advances in repertoire sequencing of T cells sorted with increasing granularity using cell surface markers (*Oakes et al., 2017*; *Soto et al., 2020*) along with advances in single-cell technologies linking TCR sequencing and cellular phenotyping could help differentiate between these scenarios in the future.

An important question raised by our work is which antigens drive the expansion of early T cell clones. To address this question, it will be necessary to determine the exposures that imprint the abundance of these clones, as has been done recently for mucosal-associated invariant T cells (*Constantinides et al., 2019*), a subset of non-conventional T cells. Going forward, the highly abundant clones with sequences close to the genetically inherited gene templates resulting from the absence of TdT expression during early fetal development are a particularly interesting target of study. They might constitute an evolutionarily controlled set of innate-like defenses within the adaptive immune system. Determining what imprints their abundances will help resolve the question of whether their large abundances are simply a byproduct of rapid repertoire formation or whether these clones serve particular functions.

## Materials and methods

### Experimental data sources

We analyzed T cell repertoire sequencing data from two large published cohort studies of healthy human volunteers by *Britanova et al., 2016* and *Emerson et al., 2017* and from a longitudinal study by *Chu et al., 2019*, detailed descriptions of which we provide in the following.

In short, *Britanova et al., 2016* sequenced reverse transcribed mRNA with added unique molecular identifiers (UMIs), while *Emerson et al., 2017* and *Chu et al., 2019* sequenced genomic DNA coding for this region without the addition of UMIs. These approaches have complementary strengths: The addition of UMIs allows to correct for stochasticity during PCR amplification and sequencing artifacts, while DNA sequencing removes the influence of cell-to-cell gene expression heterogeneity.

The Britanova cohort comprises 71 individuals spanning ages 6–103 years, as well as eight cord blood samples. The Emerson cohort spans ages 1–74 years and consists of a training and validation set of 666 and 120 individuals, respectively. From the training set, we excluded 111 samples with missing age information and 62 samples with a conflicting data format. We used only samples from the training set to analyze how the scaling law of repertoire organization changes with age (*Figure 1*). For the zero insertion enrichment analyses (*Figure 3B,C*), we combined both the training and validation set together with separately published repertoire sequencing data from eight elderly individuals *Lindau et al., 2019* generated using the same experimental pipeline (immunoSEQ, Adaptive Biotechnologies, Seattle) to achieve the broadest possible coverage of all age groups.

The longitudinal study by *Chu et al., 2019* performed repertoire sequencing of peripheral blood from three healthy female volunteers (using the immunoSEQ pipeline) over eight time points spanning a ~1 year time frame. One individual in the study was in mid-adulthood (24–45 years, Subject 3 in the original study), while two were in early adulthood (18–24 years, Subjects 1 and 2 in the original study). In *Figure 4A,B*, we display data from the older individual as we expect dynamics of large clones to be masked less by measurement noise as the large clones increase in relative abundance with age.

All studies from which we analyzed data sequenced the locus coding for the TCR CDR3 β-chain only, and we thus define clones as collections of cells sharing the same CDR3 β-chain. Clone sizes are defined as the number of distinct unique molecular identifiers (UMIs) sequenced (Britanova

**Table 1.** Repertoire sequencing data used in this study.

| Study | Link |
| --- | --- |
| *Britanova et al., 2016* | https://doi.org/10.5281/zenodo.826447 |
| *Emerson et al., 2017* | https://doi.org/10.21417/B7001Z |
| *Lindau et al., 2019* | https://doi.org/10.21417/PL2018JI |
| *Chu et al., 2019* | https://doi.org/10.21417/B7J01X |

cohort), or based on sequencing reads (Emerson cohort). The definition of a clone solely based on the CDR3 β-chain neglects convergent recombination of the most easily produced receptors with different CDR3 α-chains, but we expect convergent recombination to be sufficiently rare overall for this distinction not to qualitatively affect clone size distributions.

For all studies we used data, which was preprocessed as described in the original study. This data is publicly available using the links provided in *Table 1*.

We also used flow cytometry data on the fraction of naive cells from *Britanova et al., 2014* (available at https://doi.org/10.1371/journal.pcbi.1005572.s016) and from *Pediatric AIDS Clinical Trials Group et al., 2003*.

## Data analysis

### Fitting power-law exponents

We estimate the power-law exponent from sampled clone sizes $\{C_i\}$, $i = 1, \ldots, M$, which exceed a minimal size $C_{min}$ by numerically maximizing the log-likelihood of the data (*Clauset et al., 2009*),

$$\mathcal{L} = -M \ln \zeta(1 + \alpha, C_{min}) - (1 + \alpha) \sum_{i=1}^{M} \ln C_i, \tag{4}$$

where $\zeta(x, k)$ is the incomplete Riemann zeta function. We use $C_{min} = 16$ for both cohorts, which provides a balance between minimizing bias of the estimated exponents induced by subsampling while not overly increasing the variance of the estimator by excluding most of the data (see *Appendix 1— figure 2*).

### Fitting the zero insertion profiles

To fit the zero insertion fractions to the theory prediction (*Equation 3*) we determine the values for $r^\star$ and $\tau_d$ by a weighted least squares fit. We set $r$ and $t$ to the mid-value of each bin for the data. We weight each value by the sum of its empirical standard error and a fixed model specification error of $2 \cdot 10^{-3}$ chosen based on fits to the simulated cohort (*Figure 4—figure supplement 3*). To demonstrate the feasibility of the parameter inference, we reinferred the parameters from the simulated data and recovered those used as parameter values for the simulation. We also fitted the values of $p_{0,-}$ and $p_{0,+}$, but we note that they are not used in the rescaling and are only needed to display the theoretical curve (*Equation 3*).

### Normalization of clone sizes

Variations in sampling depth can confound comparisons of clone sizes (Appendix 1). Intuitively, if we sample more cells overall we also expect to sample proportionally more cells belonging to each given clone. This suggests to use the frequency with which cells are sampled from a given clone as a more robust measure, which can be empirically estimated by normalizing each clone size by the total sample size. We further normalize clone sizes by the fraction of memory T cells found in people of different ages to account for the increase in memory cell fraction in peripheral blood with age (Appendix 5). Together these two normalization steps lead to a large degree of data collapse as compared to unnormalized clone sizes (*Figure 1—figure supplement 3*).

### Regression analyses

We determine 95% confidence intervals on regression lines by bootstrapping using case resampling (*Efron and Hastie, 2016*).

## Mathematical framework

We describe T cell dynamics using the following general set of stochastic rate equations. The class of models we consider are known in the mathematical literature as birth-death-immigration models. The number of cells $C_i$, $i = 1, \ldots, M$ of each of the $M$ clones in the repertoire changes according to

$$\text{proliferation}: \quad C_i \xrightarrow{b_i(\boldsymbol{C},\boldsymbol{X},t)C_i} C_i + 1, \tag{5}$$

$$\text{death}: \quad C_i \xrightarrow{d_i(\boldsymbol{C},\boldsymbol{X},t)C_i} C_i - 1, \tag{6}$$

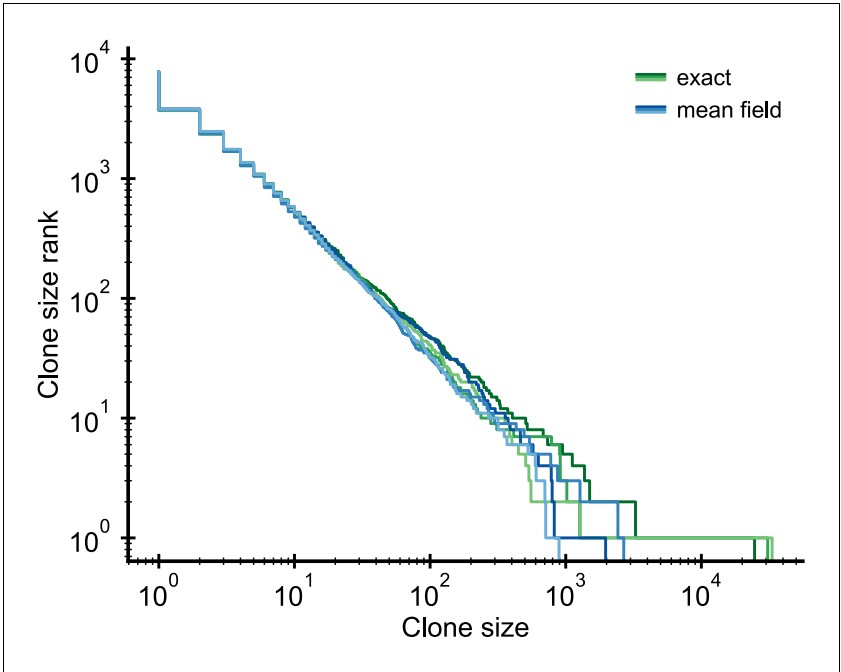

**Figure 5.** Validation of the mean-field approximation. Comparison of full stochastic simulations and simulations using mean-field competition. Parameter: $b_0 = 2 \cdot 10^4$/year, $d = 0.2$/year, $\theta = 2 \cdot 10^3$/year (implying $\gamma = 0.1$), simulation length 5 years.

where the rate of proliferation $b_i(\boldsymbol{C}, \boldsymbol{X}, t)$ or cell death $d_i(\boldsymbol{C}, \boldsymbol{X}, t)$ generally can depend on the repertoire composition $\boldsymbol{C}$, on the time $t$, and on the state of the environment $\boldsymbol{X}(t)$ representing for example the levels of different antigens and cytokines in the organism at a given time. We furthermore consider that new clones are added at rate $\theta(\boldsymbol{X}, t)$ at a size $C_0$,

$$\text{recruitment:} \quad \xrightarrow{\theta(\boldsymbol{X},t)} C_{M+1} = C_0. \tag{7}$$

This recruitment represents thymic output and antigen-driven differentiation of naive cells for the naive and memory compartment, respectively.

$$b_i(\boldsymbol{C}, \boldsymbol{X}, t) = b_0/N, \ d_i(\boldsymbol{C}, \boldsymbol{X}, t) = d, \ \theta(\boldsymbol{X}, t) = \theta \tag{8}$$

where $N(t) = \sum_{j=1}^{M(t)} C_j(t)$ is the total repertoire size. In the results section 'Long-lived incumbency advantage shows early expansions imprint clone size hierarchy' w modify this model by adding a noise term that describes the effective influence of environmental variations $\boldsymbol{X}(t)$ on clonal proliferation,

$$b_i(\boldsymbol{C}, \boldsymbol{X}, t) = b_0/N + \sqrt{2}\sigma\eta_i(t), \tag{9}$$

where $\langle \eta_i(t)\eta_j(t') \rangle = \delta_{ij}\delta(t - t')$.

## Modeling repertoire formation
### Mechanistic motivation for the competition function
We consider a population of $N$ T cells that proliferate at a rate proportional to the concentration $S$ of a set of stimuli (stimulatory cytokines), $b \propto S$. We assume that the cytokines are produced by other cells at some fixed rate $p$ and degraded at a basal rate $q$. We further assume that competition between T cells is mediated by their consumption of cytokines. The dynamics of $S$ is then described by

$$\frac{\mathrm{d}S}{\mathrm{d}t} = p - qS - kSN, \tag{10}$$

where $-kSN$ is a mass action term describing how T cells lower cytokine levels. Assuming a separation of timescales in which cytokine concentrations change quickly we obtain the quasi steady state approximation

$$S = \frac{p}{q + kN}. \tag{11}$$

When the consumption term dominates relative to basal decay, $kN \gg q$, we obtain $b \propto S \propto 1/N$.

## Mean-field competition approximation

We simplify the full stochastic model (*Equation 5–Equation 7*) using a mean-field approximation for the competition, which decouples the dynamics of individual clones while retaining the full stochasticity on the clonal level. This approximation replaces the dependence of the proliferation rate on $N$ by a dependence on its continuum theory average given by *Equation 14*. We exactly simulated a system of reduced size to validate the mean-field approximation. The distributions of the exact and mean-field simulations agree to within stochasticity (*Figure 5*), with the exception of the largest clone, which is larger in the exact simulations as has been discussed elsewhere (*Dodds et al., 2017*).

## Continuum theory of clonal growth

To obtain insight into why the model produces power-law scaling we present a simple continuum theory of early clonal dynamics. We approximate the clone size dynamics of the $i$-th clone $C_i$ as

$$\frac{dC_i}{dt} = \left( \frac{b_0}{N(t)} - d \right) C_i, \tag{12}$$

with $C_i(t_i) = C_0$ at the time of recruitment $t_i$. The total repertoire size $N = \sum_i C_i$ evolves according to

$$\frac{dN}{dt} = b_0 - dN + \theta C_0, \tag{13}$$

whose solution is given by

$$N(t) = (b_0 + \theta C_0)\left(1 - e^{-dt}\right)/d. \tag{14}$$

For times large compared to $1/d$ the total repertoire size given in *Equation 14* reaches a steady-state,

$$N_\infty = (b_0 + \theta C_0)/d, \tag{15}$$

because competition for proliferation signals acts as a homeostatic regulator. By combining *Equation 14* and *Equation 12* we derive the clonal growth law

$$C_i(t) = C_0 \left( \frac{e^{dt} - 1}{e^{dt_i} - 1} \right)^{1/(1+\gamma)} e^{-d(t-t_i)}, \tag{16}$$

where $\gamma$ as in Appendix 2 is the recruitment to proliferation ratio which in this model is given by $\gamma = \theta C_0/b_0$. To simplify we expand the growth law at leading order for small times, $t_i < t \ll 1/d$, to obtain

$$C_i(t) = C_0 \left( \frac{t}{t_i} \right)^{1/(1+\gamma)}. \tag{17}$$

This expression can also be derived directly by noting that early repertoire growth is linear $N(t) \approx (b_0 + \theta C_0)t$, and that the early dynamics is dominated by proliferation and not death such that

$$\frac{dC_i}{dt} = \frac{1}{(1+\gamma)t} C_i, \tag{18}$$

which is solved by *Equation 17*. Given the constant recruitment of new clones the distribution of the $t_i$'s is uniform, which with *Equation 17* implies a clone size distribution

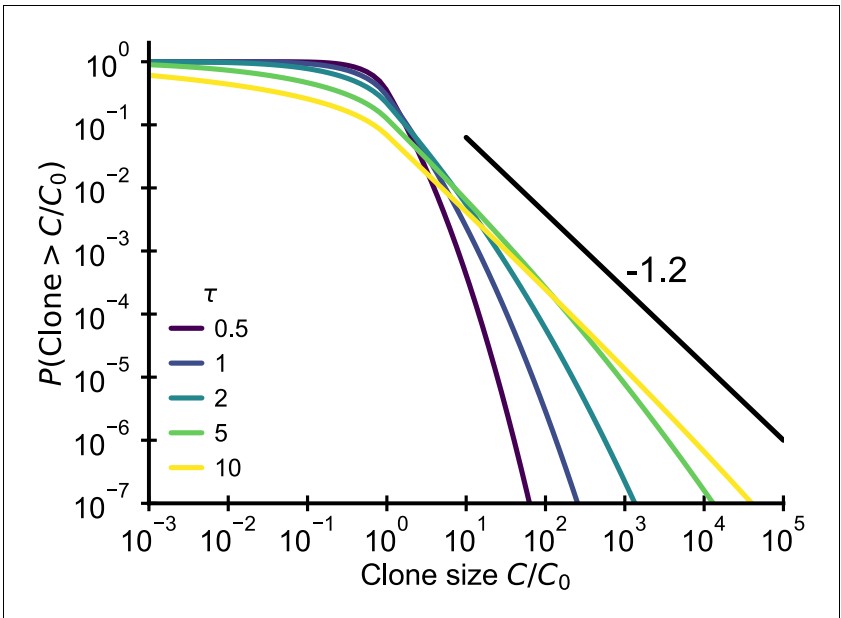

**Figure 6.** Fluctuating fitness model out-of-steady state. Analytical predictions for the clone size distributions in a geometric Brownian motion fluctuating fitness model (Integral of *Equation 26*) as a function of effective age $\tau = T\sigma^2$. The black line shows the asymptotic prediction for the steady-state scaling. Parameter: $\alpha = 1.2$.

$$P(C) = P(t_i(C)) \left| \frac{dt_i}{dC} \right| \propto C^{-2-\gamma} \tag{19}$$

that follows power-law scaling with an adjustable exponent that depends on $\gamma$. Note that the exponent for $P(C)$ differs by one from the exponent for the rank (*Clauset et al., 2009*), which is a complementary cumulative distribution, and thus $\alpha = 1 + \gamma$.

### Steady-state distribution

To derive the power-law scaling, we have expanded the total repertoire size for small times (or death rates). How does the clone size distribution change later in life? At large times, the division rate $b_0/N(t)$ falls below the constant death rate $d$ as the steady-state repertoire size $N_\infty$ is approached following *Equation 14*. In this model this happens at a time $t^\star \simeq \log(1 + 1/\gamma)/d$, after which the large clones experience a deterministic force toward extinction. For times $t \gg t^\star$ the model effectively reduces to the neutral birth-death dynamics considered in Appendix 2. (The growth rate fluctuations produced by variations of the total population size around steady state asymptotically vanish for large $N_\infty$.) We thus expect the steady-state clone size distribution to be equivalent to that of the neutral model (*Equation 48*). Indeed this distribution accurately describes the distribution of small clones in old age (*Figure 2B*). The neutral distribution is not compatible with data as discussed before. However, the timescale over which large early founded clones vanish is long (Appendix 2) such that a tail of large clones resulting from the early growth dynamics can be maintained much beyond $t^\star$ until $t \gg \tau_c$ (see also Appendix 2 –Relaxation time scale in a neutral model).

## Modeling long-term repertoire dynamics with fluctuating clonal growth rates

### Slow convergence to steady-state scaling

Multiplicative stochastic processes are a classical generative mechanisms for heavy-tailed distributions (*Sornette and Cont, 1997*; *Gabaix, 1999*; *Newman, 2005*). In the context of lymphocyte dynamics this mechanism has first been proposed by *Desponds et al., 2016*, who argued that fluctuations in antigen availability can lead to multiplicative stochastic dynamics producing power-law scaling at steady state. Here, we expand on this earlier work by analyzing a simple fluctuating fitness

model out-of-steady-state. Our analytical results show that the emergence of scaling can be slow when the fluctuation amplitude is small.

We opted to treat proliferation rate fluctuations as temporally uncorrelated for computational tractability (*Equation 2*). Correlations in proliferation rate fluctuations are clearly an important feature of short-term dynamics – for example to describe the quick expansion and contraction during and following acute infection over a timescales of days and weeks, respectively (*Mayer et al., 2019*). However, given finite correlation times we expect to be able to capture dynamics over the long timescales which we are interested in here, with uncorrelated noise with an effective net fluctuation strength that averages over the short-term dynamics.

In this limit clone sizes follow a geometric Brownian motion, that is $x = \log C/C_0$ follows the Langevin equation

$$\frac{\mathrm{d}x_i}{\mathrm{d}t} = f_0 + \sqrt{2}\sigma\eta_i, \tag{20}$$

with initial condition $x(t_i) = 0$, where $\sigma$ sets the fluctuation strength and where $\langle\eta_i(t)\eta_j(t')\rangle = \delta_{ij}\delta(t - t')$. A negative mean fitness $f_0 < 0$ balances the recruitment of new clones and the net expansion induced by the fluctuating term. In general, we might want to include also demographic noise and the extinction of clones as an absorbing boundary condition (*Desponds et al., 2016*), but here for simplicity we will neglect those effects. *Equation 20* is a diffusion equation for the logarithmic clone size $x$ and has the well-known Green's function

$$G(x,y,t) = \frac{1}{\sqrt{4\pi\sigma^2 t}}e^{-\frac{(x-y-f_0 t)^2}{4\sigma^2 t}}, \tag{21}$$

which describes how the distribution spreads out from an initial $\delta$-distribution centered at size $y$. The clone size distribution at time $T$ is given by

$$P(x,T) = \int_0^T \mathrm{d}t P(t)G(x,0,t), \tag{22}$$

where $t$ is the clonal age. For a constant immigration rate $t$ is uniformly distributed and we obtain by integration

$$P(x,T) = \frac{e^{\frac{f_0 x(1-\theta(x))}{\sigma^2}}\mathrm{erfc}\left(\frac{|x|-f_0 T}{\sqrt{4T\sigma^2}}\right) - e^{\frac{f_0 x\theta(x)}{\sigma^2}}\mathrm{erfc}\left(\frac{|x|+f_0 T}{\sqrt{4T\sigma^2}}\right)}{2f_0 T}, \tag{23}$$

where $\theta(x)$ is the Heaviside step function, $\theta(x) = 0$ for $x < 0$ and $\theta(x) = 1$ otherwise. For large $T$ and $x > 0$ this reduces to

$$P(x) \to e^{\frac{f_0 x}{\sigma^2}}/(-f_0 T), \tag{24}$$

which implies

$$P(C) \sim C^{-(1+\alpha)} \quad \text{with } \alpha = -f_0/\sigma^2, \tag{25}$$

recovering the steady-state result from *Desponds et al., 2016*.

Setting $f_0 = -\alpha\sigma^2$ and rescaling age as $\tau = T\sigma^2$, we can rewrite the finite time solution as

$$P(x,\tau) = \frac{e^{-\alpha x\theta(x)}\mathrm{erfc}\left(\frac{|x|+\tau\alpha}{\sqrt{4\tau}}\right) - e^{-\alpha x(1-\theta(x))}\mathrm{erfc}\left(\frac{|x|-\alpha\tau}{\sqrt{4\tau}}\right)}{2\alpha\tau}. \tag{26}$$

Plotting the cumulative distribution of clone sizes at different effective ages (*Figure 6*) we observe that the convergence of clone size distributions is slow when $\sigma^2$ is small. Based on estimates for the fluctuation strength from longitudinal data (*Figure 4B*), we would expect significant deviations from the steady state power-law scaling that persist into adulthood. Thus, this mechanism alone is unable to account for the observed power-law scaling in data.

## A note on the scaling exponent

A minimal requirement for the existence of a steady state is $f_0<0$ ensuring that clones eventually die to balance the recruitment of new clones. This condition still allows such multiplicative processes to produce power-laws with arbitrary exponents as noted before (**Desponds et al., 2016**). Here, we propose that the parameters should fulfill a stronger condition. In particular, it seems reasonable to require that the large clones do not deterministically take up a larger fraction of the overall repertoire, or equivalently that their expected change in clone size should not exceed one. The mean of the lognormal distribution of clone size change is given by $e^{f_0+\sigma^2}$, and thus we find the stronger condition

$$-f_0<\sigma^2. \tag{27}$$

Importantly, it follows that exponents in the vicinity of $\alpha=-1$ arise without fine-tuning as long as the timescale of expected net clonal decay is large compared to the diffusion timescale.

Another perspective on the parameterization is provided by noting that the Langevin equation for $C$ (not $x=\log C$) in the Stratonovich convention includes an extra drift term $-\sigma^2$, to keep $\langle\Delta C\rangle$ independent of the choice of $\sigma$. Alternatively, in the Ito convention the extra drift term arises by Ito's lemma when transforming the equation from $C$ to $x$.

## Predictions for longitudinal fluctuations in clone sizes

To quantify longitudinal fluctuations, we calculate the mean and variance of log-clonesize changes with respect to a reference time $t_0$. From the model we, according to **Equation 21,** expect

$$\langle x(t)-x(t_0)\rangle=f_0t \tag{28}$$

$$\langle(x(t)-x(t_0)-\langle x(t)-x(t_0)\rangle)^2\rangle=2\sigma^2t. \tag{29}$$

The variance of log-clonesize changes in empirical data involves an additional term $\sigma_S^2$ accounting for sample-to-sample variability. This term is expected not to depend on the time difference, and we can thus determine $\sigma^2$ by linear regression with an intercept that captures the sampling variability $\sigma_S^2$ (**Figure 4B**).

We note that a similar approach has been independently proposed in unpublished work by **Ferri, 2018**.

## Relaxation of the zero insertion distribution

Here, we solve for the relaxation dynamics of the zero insertion distribution in a simplified setting. Throughout we use log clone sizes $x=\log C$ for notational convenience. We posit that at time 0 the power-law distribution $P(x,0)=\alpha e^{-\alpha x}$ is already established and we further assume that the $r^\star$ largest clones have zero insertion probability $p_{0,-}$ and all smaller or later added clones have probability $p_{0,+}$. Then the probability that a clone of a given size $x$ has zero insertions is given by

$$P_0(x,t)=\Delta p_0 f_{early}(x,t)+p_{0,+} \tag{30}$$

where $\Delta p_0=p_{0,-}-p_{0,+}$ and $f_{early}(x,t)$ is the fraction of clones of size $x$ and time $t$ that derive from the $r^\star$ largest clones at time 0.

In the following, we determine an analytical formula for $f_{early}(x,t)$ under the assumption that the dynamics leaves the distribution unchanged $P(x,t)=P(x,0)$. We then have

$$f_{early}(x,t)=\frac{\int_{x_{min}}^\infty \mathrm{d}y\, e^{-\alpha y}G(x,y,t)}{e^{-\alpha x}}, \tag{31}$$

where $G(x,y,t)$ as before is the Green's function of the fluctuating proliferation rate dynamics and $x_{min}$ is defined such that the total number of clones times $P(x>x_{min})$ equals $r^\star$. By integration one obtains

**Table 2.** Parameters of the minimal model of repertoire formation (*Figure 2B*).

| Parameter | Explanation | Value |
|---|---|---|
| $d$ | death rate | 0.2/year |
| $\gamma$ | recruitment to proliferation ratio | 0.1 |
| $\theta$ | recruitment rate | $10^6$/year |
| $C_0$ | recruitment size | 1 |

$$f_{early}(x,t) = \frac{1}{2}e^{\alpha t(f_0 + \alpha\sigma^2)}\operatorname{erfc}\left(\frac{x_{min} - x + t(f_0 + 2\alpha\sigma^2)}{\sqrt{4\sigma^2 t}}\right), \tag{32}$$

which after setting $f_0 = -\alpha\sigma^2$ reduces to

$$f_{early}(x,t) = \frac{1}{2}\operatorname{erfc}\left(\frac{x_{min} - x + \alpha\sigma^2 t}{\sqrt{4\sigma^2 t}}\right). \tag{33}$$

To convert clone size into ranks, we note that $\text{rank} \sim e^{-\alpha x}$ and thus $x_{min} - x \sim \frac{1}{\alpha}\log\left(\frac{r}{r^\star}\right)$. In combination with *Equation 33* and *Equation 30* we thus obtain

$$P_0(r,t) = \frac{\Delta p_0}{2}\operatorname{erfc}\left(\frac{\frac{1}{\alpha}\log(r/r^\star) + \alpha\sigma^2 t}{\sqrt{4\sigma^2 t}}\right) + p_{0,+}. \tag{34}$$

Defining a characteristic timescale for the diffusive dynamics as $\tau_d = 1/(\alpha\sigma)^2$, we can simplify this expression to

$$P_0(r,t) = \frac{\Delta p_0}{2}\operatorname{erfc}\left(\frac{\log(r/r^\star) + t/\tau_d}{2\sqrt{t/\tau_d}}\right) + p_{0,+}. \tag{35}$$

## Simulation procedures

### Repertoire formation

To simulate the model efficiently at large scales, we use a mean-field competition approximation (Materials and methods – Mean–field competition approximation). We verified the validity of the mean-field assumption by comparing them to full stochastic simulations of the coupled birth-death-immigration equations, which we simulated using the Gillespie algorithm (*Press et al., 2007*; *Figure 5*). In the mean-field approximation, the proliferation rate is time-dependent, which requires a specific procedure for sampling event times. The time interval until the next event depends on the total rate for all possible processes $\lambda(t) = \theta + b(t) + d$. To sample an interval of time $\Delta t$ between two events from an inhomogeneous Poisson process of rate $\lambda(t)$ one can sample from a Poisson process with a rate function $\lambda^\star(t)$ fulfilling the majoration condition $\lambda^\star \geq \lambda(t)\,\forall t$ and then reject a proposed time interval $\Delta t^\star$ with a probability of $1 - \lambda(t + \Delta t^\star)/\lambda^\star(t + \Delta t^\star)$ (*Lewis and Shedler, 1979*). The thinned set of event times follows the statistics of the Poisson process with rate $\lambda(t)$. Here, because competition is increasing with time, $\lambda(t)$ decreases monotonically. Therefore, the homogeneous Poisson process with a constant rate function $\lambda^\star(t) = \lambda(t_0)$, satisfies the majoration condition. Using this thinning technique, we are able to efficiently sample the next event time while accounting for the time-dependence of the proliferation rate. The source code that allows reproduction of the statistical analyses and numerical results reported in the manuscript is published (*Gaimann et al., 2020*).

### Simulated cohort

As empirical evidence shows that the tail of the clone size distribution is almost exclusively driven by cells with memory phenotype (Appendix 5), we focused on the clone size dynamics within the memory compartment. We assumed that the recruitment size for memory cells is independent of the prior naive cell dynamics, and we thus did not explicitly model the clone size dynamics within the naive compartment. Within the memory compartment, we modeled clone size dynamics under the combined effect of early deterministic expansions during repertoire formation and fluctuating clonal growth rates according to *Equation 2*. Given the large sizes of memory clones, we expect

demographic stochasticity to be negligible relative to clone size variability introduced by fluctuating selection. For tractability, we thus ignored demographic fluctuations, which allowed us to combine the continuum solution to the deterministic clonal growth (*Equation 16*) with the stochastic propagator for the fluctuating dynamics (*Equation 21*) to efficiently simulate the dynamics. To study the enrichment of zero insertion clones in silico, we assigned newly recruited memory clones as having zero insertions with a probability equal to the fraction $p_0(t)$ of zero insertion clones within the naive compartment. We assumed $p_0(t) = p_{0,-}$ before TdT expression turn-on at time $t^\dagger$ and $p_0(t) = p_{0,-}t/t^\dagger + p_{0,+}(1 - t/t^\dagger)$ for $t > t^\dagger$, where $t/t^\dagger$ is the fraction of naive clones produced since the switch to the adult recombination statistics. Taken together, these simplifications lead to the following direct sampling scheme:

- Sample the age $T$ of an individual uniformly from the range $[0, 80]$ years.
- Set the number of clones equal to $\theta T$ (rounded to the nearest integer), where $\theta$ is the rate of recruitment of new clones to the memory compartment.
- For each clone determine its recruitment time $t_i$ by drawing uniformly from the range $[0, T]$.
- Assign each clone as having zero insertions with a probability

$$p_0(t) = \begin{cases} p_{0,-} & t < t^\dagger \\ p_{0,-}t/t^\dagger + p_{0,+}(1 - t/t^\dagger) & \text{otherwise} \end{cases}$$

- Sample the size $C_i(T)$ of each clone as follows (*Equation 16* and *Equation 21*),

$$C_i = \exp(x_i), \quad x_i \sim \mathcal{N}\left(-d(T - t_i) + \frac{1}{1 + \gamma}\log\left(\frac{e^{dT} - 1}{e^{dt_i} - 1}\right) - \sigma^2(T - t_i), 2\sigma^2(T - t_i)\right), \quad (36)$$

where $d, \gamma, \sigma^2$ are model parameters and $y \sim N(\mu, \sigma^2)$ indicates x being drawn from a normal distribution of mean μ and variance $\sigma^2$.

- Finally to mimic the experimental sampling depth of $N_{sample}$ reads we determine sampled clone sizes $\tilde{C}_i$ by Poisson sampling,

$$\tilde{C}_i \sim Pois(N_{sample} \cdot C_i/N), \quad \text{with } N = \sum_i C_i, \quad (37)$$

where $x \sim \text{Pois}(\lambda)$ indicates x being drawn from a Poisson distribution of parameter $\lambda$.

## Parameter choices

In the following, we summarize the parameters we used to simulate repertoire dynamics and provide additional motivation for our parameter choices.

Lifetimes of several years and several months have been measured by deuterium labeling for naive and memory T cells, respectively (*De Boer and Perelson, 2013*; *Borghans et al., 2018*). Clonal turnover can be substantially slower than cellular turnover when proliferation balances most death (Appendix 2). This has been shown to be the case for the maintenance of naive cells in human (*Zhang et al., 2014*), where the aging-associated decline of the fraction of T cells with TCR excision

**Table 3.** Parameters of the simulated cohort (*Figure 3C*).

| Parameter | Explanation | Value |
| --- | --- | --- |
| $\sigma^2$ | Magnitude of clone size fluctuations | 0.08/year |
| $d$ | Death rate | 0.2/year |
| $\gamma$ | Recruitment to proliferation ratio | 0.1 |
| $\theta$ | Recruitment rate | $10^5$/year |
| $p_{0,-}$ | Zero insertion fraction early in life | 0.07 |
| $p_{0,+}$ | Adult zero insertion fraction | 0.02 |
| $t^\dagger$ | Time of recombination statistics switch | 0.05 years |
| $N_{sample}$ | Simulated sample size | $5 \cdot 10^5$ |

circles (TRECs) suggests $\gamma \sim 0.1$. Similarly, memory T cell numbers decline much more slowly overall than suggested by the deuterium labeling literature, which is thought to be driven by homeostatic proliferation in the absence of reinfection (*Macallan et al., 2017*). For example, T cell memory has been observed to decline with half-lives of 8–15 years by following titers after small pox vaccination (*Hammarlund et al., 2003*). Additionally, the relatively short average lifetime of memory T cells likely masks substantial heterogeneity with a subset of more long-lived cells also contributing to the slower long-term decline of memory cells (*Akondy et al., 2017*). Another line of direct evidence for long clonal persistence has come from two studies of identical twins (*Pogorelyy et al., 2017*; *Tanno et al., 2020*), which have shown an excess sharing of identical clones decades after in utero blood exchange in monochorionic twins.

To simulate repertoire formation (*Figure 2B*), we used a set of parameters summarized in *Table 2*. In choosing these parameters, we were not trying to reproduce exact values of any particular T cell subset, but rather illustrate a plausible biological parameter regime that characterizes T cell dynamics. We note that qualitatively the scaling presented in *Figure 2B* only depends on $\gamma$ as shown by our theoretical analysis, in a way that is shown in *Figure 2D*. For the recruitment rate $\theta$, we used a rate intermediate between those suggested by estimates of thymic output and the rate of recruitment of memory clones suggested by estimates of the diversity of the memory compartment (see below). We note that under mean-field competition the rate of recruitment $\theta$ only determines the overall number of clones, but does not influence the dynamics of individual clones. While $\theta$ decreases during adulthood for both the naive compartment (as thymic production wanes with age) and the memory compartment (as new primary responses are rarer at advanced age), we used a constant $\theta$ for simplicity. We expect this simplifying assumption not to qualitatively affect the results in *Figure 2* due to a separation of timescales: The clone size scaling emerging from the expansionary dynamics only depends on the rate $\theta$ during infancy and not on slower changes in $\theta$ happening during adulthood. We chose a recruitment size of one to numerically investigate potential deviations from the continuum theory predictions due to demographic stochasticity. The number of recruited clones is of course much larger in practice in the memory compartment, and even naive cells undergo a few rounds of division before thymic export and thus have $C_0 > 1$. Assuming a larger $C_0$ will further decrease the small effects of demographic stochasticity relative to the continuum theory.

To study the enrichment of zero insertion clones in a simulated cohort (*Figure 4C*), we used the same recruitment to proliferation ratio and death rate as in the previous simulation of repertoire formation. To determine the absolute number of large clones that have zero insertions in these simulations, the choice of the recruitment rate $\theta$ is important. Based on order-of-magnitude estimates of the clonal diversity of the memory compartment (*Robins et al., 2009*; *Qi et al., 2014*), we chose a value of $\theta = 10^5$/year. Additionally, we chose a fraction of zero insertion clones within the early naive compartment of $p_{0,-} = 0.07$ (roughly equal to their overall fraction in cord blood [*Pogorelyy et al., 2017*]) and in the late naive compartment equal to $p_{0,+} = 0.02$ (roughly equal to their overall fraction in adult blood). Finally, we used $t^\dagger = 0.05$ years for the time of the recombination switch, which together with the choice of $\theta$ produces $\sim 10^4$ excess zero insertion clones recruited during repertoire formation in line with the empirical enrichment data in the <10 years age group (*Figure 3B*). All parameters are summarized in *Table 3*.

## Acknowledgements

We thank William Bialek, Curtis Callan, Ivana Cvijovic, Yuval Elhanati, Simone Mayer, Mikhail Pogorelyy, and Ned Wingreen for discussions and comments on the manuscript. This work was supported by a DAAD RISE Worldwide fellowship (MUG), the NSF-Simons Center for Quantitative Biology under grants Simons Foundation SFARI/597491-RWC and National Science Foundation 17764421 (JD), and a Lewis–Sigler fellowship (AM).

## Additional information

### Funding

| Funder | Grant reference number | Author |
|---|---|---|
| Princeton University | Lewis-Sigler fellowship | Andreas Mayer |
| Deutscher Akademischer Austauschdienst | RISE fellowship | Mario U Gaimann |
| Simons Foundation | SFARI/597491-RWC | Jonathan Desponds |
| National Science Foundation | 1764421 | Jonathan Desponds |

The funders had no role in study design, data collection and interpretation, or the decision to submit the work for publication.

### Author contributions

Mario U Gaimann, Software, Formal analysis, Writing - original draft, Writing - review and editing; Maximilian Nguyen, Jonathan Desponds, Software, Formal analysis, Writing - review and editing; Andreas Mayer, Conceptualization, Data curation, Software, Formal analysis, Supervision, Writing - original draft, Writing - review and editing

### Author ORCIDs

Mario U Gaimann (iD) https://orcid.org/0000-0002-2789-090X
Maximilian Nguyen (iD) https://orcid.org/0000-0002-4378-5050
Jonathan Desponds (iD) https://orcid.org/0000-0001-7112-3217
Andreas Mayer (iD) https://orcid.org/0000-0002-6643-7622

### Decision letter and Author response

Decision letter https://doi.org/10.7554/eLife.61639.sa1
Author response https://doi.org/10.7554/eLife.61639.sa2

## Additional files

### Supplementary files

• Transparent reporting form

### Data availability

No new data was generated in this study. All source codes associated with this manuscript are available online at https://github.com/andim/paper-tcellimprint (copy archived at https://archive.softwareheritage.org/swh:1:rev:9029ffeeb645d02f1fc880a89e136448c6430f49/).

The following previously published datasets were used:

| Author(s) | Year | Dataset title | Dataset URL | Database and Identifier |
|---|---|---|---|---|
| Britanova OV, Shugay M, Merzlyak EM, Staroverov DB, Putintseva EV, Turchaninova MA, Mamedov IZ, Pogorelyy MV, Bolotin DA, Izraelson M, Davydov AN, Egorov ES, Kasatskaya SA, Rebrikov DV, Lukyanov S, Chudakov DM | 2016 | Dynamics of individual T cell repertoires: from cord blood to centenarians | https://doi.org/10.5281/zenodo.826447 | Zenodo, 10.5281/zenodo.826447 |

| Emerson RO, DeWitt WS, Vignali M, Gravley J, Hu JK, Osborne EJ, Desmarais C, Klinger M, Carlson CS, Hansen JA, Rieder M, Robins HS | 2017 | Immunosequencing identifies signatures of cytomegalovirus exposure history and HLA-mediated effects on the T cell repertoire | https://doi.org/10.21417/B7001Z | immunoACCESS, 10.21417/B7001Z |
| Chu ND, Bi HS, Emerson RO, Sherwood AM, Birnbaum ME, Robins HS | 2019 | Longitudinal immunosequencing in healthy people reveals persistent T cell receptors rich in highly public receptors | https://doi.org/10.21417/B7J01X | immunoACCESS, 10.21417/B7J01X |
| Lindau P, Mukherjee R, Gutschow MV, Vignali M, Warren EH, Riddell SR, Makar KW, Turtle CJ, Robins HS | 2019 | Cytomegalovirus Exposure in the Elderly Does Not Reduce CD8 T Cell Repertoire Diversity | https://doi.org/10.21417/PL2018JI | immunoACCESS, 10.21417/PL2018JI |

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

## Appendix 1

### Subsampling scaling

Only a small fraction of the ~$10^{12}$ T cells in the human body are sampled by repertoire sequencing. What effect does subsampling have on the clone size distribution? In the following, we discuss how subsampling affects the distribution of sampled clone sizes and we discuss analysis techniques for robust inferences and data visualization despite variations in sampling depth.

### Inference of scaling exponent

Given a clone of size $C$ in the repertoire, the number of reads from that clone $\tilde{C}$ follows a distribution $P(\tilde{C}|C)$. The form of $P(\tilde{C}|C)$ depends on the sampling process. To build intuition let us consider the simplest case, in which every cell is sampled independently with a probability $\eta$, the subsampling fraction. Then the sampling distribution is binomial

$$P(\tilde{C}|C) = \binom{C}{\tilde{C}} \eta^{\tilde{C}} (1-\eta)^{C-\tilde{C}}. \tag{38}$$

The mean of this distribution is

$$\langle \tilde{C} \rangle = \eta C, \tag{39}$$

which implies that sampled clone sizes are on average smaller by a factor $\eta$ than the actual clone size. In the practically relevant limit where the sampling fraction is small, $\eta \ll 1$, we can further simplify and assume that the counts from the large clones follow a Poisson distribution. In the Poisson limit, the sampled clone size varies around its mean value with a coefficient of variation that scales as an inverse of the square root of the mean sampled count,

$$c_v = \frac{\sqrt{\langle (\tilde{C} - \langle \tilde{C} \rangle)^2 \rangle}}{\langle \tilde{C} \rangle} = \frac{1}{\sqrt{\eta C}}. \tag{40}$$

Importantly, the stochastic sampling introduces a subsampling scale, $\tilde{C} = \eta C \sim 1$, at the clone size $C = 1/\eta$, from which on average we expect a single sampled cell. Due to the existence of this scale subsampling breaks scale-invariance: even if $P(C)$ follows a perfect power law, the distribution of sampled counts

$$P(\tilde{C}) = \sum_C P(C) P(\tilde{C}|C) \tag{41}$$

deviates from power-law scaling close to $\tilde{C} = 1$. This intuition can be made rigorous using a generating function formalism (*Stumpf et al., 2005*): for example for $P(C) = C^{-2}/\zeta(2)$ one obtains for $\tilde{C} > 1$

$$P(\tilde{C}) \sim \frac{1}{\tilde{C}(\tilde{C}-1)}. \tag{42}$$

As expected the scaling with an exponent $-2$ is recovered asymptotically, but subsampling leads to a deviation from scaling when $\tilde{C}$ is close to 1.

The deviation from scaling due to subsampling leads to biases in naive estimates of the scaling exponent. How can we determine a power-law exponent in a way that is robust to subsampling? When the sampling distribution is known or can be inferred from replicate sequencing the exponent can be determined using maximum likelihood estimation of a model with an underlying power law distribution of clone sizes convolved with the sampling probability (*Puelma Touzel et al., 2020*). Here, we propose a simpler approach that does not require precise knowledge of the sampling process. We exploit the fact that the deviations from scaling vanish asymptotically for large $\tilde{C}$ (*Equation 42*), by excluding small clones below some minimal size $C_{min}$ from the fitting. The power-law exponent is expected to converge as we increase $C_{min}$, which we confirm using simulated data (*Appendix 1—figure 1*, blue dots). We can also consider more realistic models for the sampling

process that account for overdispersion, that is their coefficient of variation exceeds the minimal value of one set by Poisson sampling. Mechanistically, such overdispersion arises for a number of reasons, most importantly because in practice we are not actually directly counting cells: in the DNA-based sequencing pipeline every cell can give rise to multiple sequencing reads due to the PCR amplification step, and in the mRNA-based sequencing pipeline despite the addition of unique molecular identifiers several of them can originate from different mRNA molecules from the same cell. As long as the number of reads from each cell is independently and identically distributed the law of large numbers ensures that the relative frequencies of large clones converge. We thus expect that the trimming method of fitting only to counts greater than $C_{min}$ also works for overdispersed sampling. We test the trimming method on simulated data, in which the sampling follows a negative binomial distribution with mean $\mu$ and variance $\mu + a\mu^2$ (which reduces to Poisson sampling for $a = 0$). We find that trimming allows a correct estimate of $\alpha$ (*Appendix 1—figure 1*, orange and green dots).

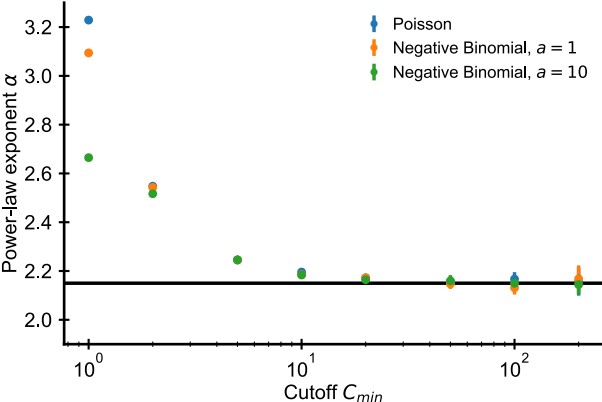

**Appendix 1—figure 1.** Estimated power-law exponents converge to correct value using trimming method. Fitted exponent as a function of the cutoff choice in simulated data (errorbars ±2 SE over 50 independent draws). The fitted exponent changes drastically for small $C_{min}$ before leveling off indicating deviations from true power-law scaling at the smallest clone sizes. Such a deviation is expected due to subsampling despite the true power-law scaling in the underlying distribution (see text). Simulations: $10^7$ clones were drawn from a discrete power-law distribution with $\alpha = 2.15$. A sample of size $5 \cdot 10^5$ cells was then drawn from the underlying power law based on a Poisson (blue dots) or negative binomial sampling (orange and green dots show two choices of the overdispersion coefficient $a$).

Applying the same method to the empirical data we find that the fitted exponents also depend on $C_{min}$ (*Appendix 1—figure 2*). In practice, we chose $C_{min} = 16$ to balance a trade-off between minimizing bias and variance, which increases as more of the data is excluded from the fit *Appendix 1—figure 2*.

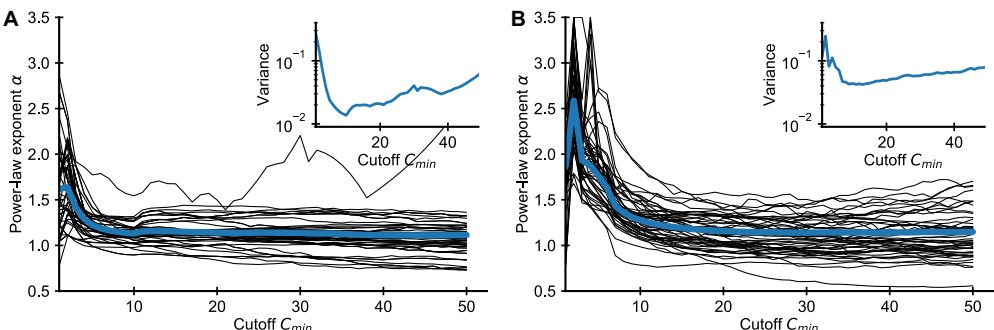

*Appendix 1—figure 2 continued on next page*

**Appendix 1—figure 2.** Influence of choice of $C_{min}$ on fitted power-law exponent for empirical data. Fitted exponent as a function of the cutoff choice (black lines: 50 random repertoires, blue line: mean) in the (**A**) Britanova et al. and (**B**) Emerson et al. datasets. The fitted exponent changes drastically for small $C_{min}$ before leveling off indicating deviations from true power-law scaling at the smallest clone sizes, similarly to those seen in simulated data (*Figure 1*). To alleviate the bias induced by finite sampling, we choose a cutoff value $C_{min}$, for which the power-law exponent estimates have leveled off. For large $C_{min}$ the variance of fitted exponent increases as more and more data is excluded from the fit (A, B inset), which sets a practical upper bound for choosing $C_{min}$.

## Graphical display of subsampled distributions

The intuition we have built about how subsampling affects clone size distributions can help us choose an appropriate method for displaying subsampled data (*Appendix 1—figure 3*). Which graphical representation of the clone size distribution minimizes the influence of variations in sampling depth?

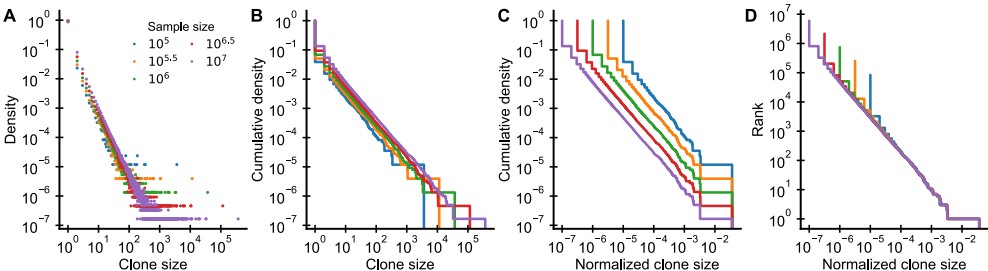

**Appendix 1—figure 3.** Graphical display of subsampled power-law distributions. (**A-D**) Show various ways of displaying clone size distributions obtained by subsampling an underlying clone size distribution consisting of $10^8$ clones drawn according to $P(C) \sim C^{-2.2}$ to various sampling depths. (**A**) The empirical probability density function of clone sizes, (**B**) its cumulative density, as well as (**C**) the cumulative density of normalized clone sizes are not invariant under changes of the sampling depth. Only the tail behavior of relative frequencies of finding cells from large clones is reproducibly captured, which makes rank-frequency plots (displays of unnormalized cumulative distributions of normalized clone sizes) the method of choice for collapsing clone size distributions at various sampling depths.

The shift of the mean clone size (*Equation 39*) suggests that we should normalize sampled clone sizes by the sampling fraction $\eta$, as has been noted elsewhere (*Levina and Priesemann, 2017*). While experimentally we do not know the sampling fraction, we can instead simply divide the clone sizes by the total sample size (*Appendix 1—figure 3C,D*). This normalization is particularly intuitive as it corresponds to using the relative frequencies of cells in different clones. While the absolute number of cells in a large clone increases with more sampling, the fraction of all sampled cells that are part of a particular clone remains constant on average.

Plots of the cumulative distribution of clone sizes make it easier to visually assess the tail behavior of the distribution (*Appendix 1—figure 3B*) than plots of the probability density (*Appendix 1—figure 3A*). However, even after normalizing clone sizes by the sample size there remains a very visible shift between the cumulative distributions at different sampling depths (*Appendix 1—figure 3C*). This shift arises because the implicit normalization by the total number of unique clones makes the sampled cumulative distribution depend heavily on sampling depth. As sampling increases so does the total number of unique clones that will be sequenced. This suggests that we might do better by simply omitting the normalization. Ranking clones by their normalized size yields precisely such an unnormalized cumulative distribution. Taken together, by both scaling clone sizes by the sample size and resisting the temptation to normalize the ranks, we can collapse distributions sampled at different depths (*Appendix 1—figure 3D*).

## Appendix 2

### Modeling neutral repertoire dynamics

In the following, we review results on the neutral dynamics of clone sizes in which the continuous recruitment of new clones is balanced by a net negative growth of already established clones $b<d$. These models have a long history in ecology (*Volkov et al., 2003*), and have also been proposed as null models in the context of T cell dynamics previously (*Desponds et al., 2017*; *Desponds et al., 2016*; *de Greef et al., 2020*). While the results can be found in the literature their inclusion serves to introduce a parametrization highlighting the recruitment to proliferation ratio $\gamma$ as a key quantity governing clonal dynamics.

#### Steady-state clone size distribution

At steady state, the probability distribution $P(C)$ of clone sizes $C$ needs to fulfill the balance condition

$$bCP(C) = d(C+1)P(C+1), \tag{43}$$

for all $C>C_0$ which yields

$$P(C) \propto \frac{1}{C}\left(\frac{b}{d}\right)^C = \frac{1}{C}\exp(-C\log(d/b)). \tag{44}$$

This distribution is characterized by power-law scaling with an exponent of 1 for small clone sizes, and, importantly, has an exponential cutoff at $C^\star = 1/\log(d/b)$. In contradiction with this model experiments point toward power-law scaling with an exponent ~2 (Note that $P(C) \sim C^{-\alpha-1}$ when $\mathrm{rank} \sim C^{-\alpha}$). Additionally, the size of large clones seen experimentally is incompatible with the predicted exponential cutoff as we discuss below.

The total repertoire size follows the following continuum equation

$$\frac{\mathrm{d}N}{\mathrm{d}t} = (b-d)N + \theta C_0, \tag{45}$$

such that at steady-state, $\frac{\mathrm{d}N}{\mathrm{d}t} = 0$, the repertoire has a total size

$$N_\infty = \frac{\theta C_0}{d-b}. \tag{46}$$

For a more interpretable alternative parametrization, we introduce the recruitment to proliferation ratio for the maintenance of cells at steady state

$$\gamma = \frac{\theta C_0}{bN_\infty} = \frac{d}{b} - 1. \tag{47}$$

Using this relation to rewrite *Equation 44* we obtain

$$P(C) \propto \frac{1}{C}\exp(-C\log(1+\gamma)), \tag{48}$$

implying a cutoff clone size of $C^\star = 1/\log(1+\gamma)$. The largest clones represent on the order of one percent of the repertoire, which assuming independent sampling from the underlying repertoire would correspond to ~$10^{10}$ cells in the complete repertoire. For small $\gamma$, we can expand $C^\star \approx 1/\gamma$, so in order to have a cutoff clone size $C^\star$ of this order of magnitude one would need to have an unreasonably small $\gamma \sim 10^{-10}$.

#### Relaxation time scale in a neutral model

Over what timescale do transiently expanded clones disappear in a neutral model? The time scale $\tau_c = \frac{1}{d-b}$ for deterministic clonal decay can be much larger than the lifespan $1/d$ of a single cell when birth and death are closely balanced. Rewriting the birth rate in terms of $\gamma$ and $d$ we obtain

$$\tau_c = \frac{1+\gamma}{d\,\gamma},\qquad(49)$$

demonstrating that for $\gamma \ll 1$ clonal dynamics is a factor of $1/\gamma$ slower than cellular dynamics.

## Appendix 3

### Influence of model assumptions on repertoire formation process

For tractability and interpretability, we have kept the model presented for repertoire formation in the main text deliberately simple. Here, we explore how a saturation of the proliferation rate, competition for specific resources, or variations in the recruited number of cells impact the clone size distributions.

#### Saturation of proliferation rate

Cellular growth is not arbitrarily fast, which is not accounted for in the simple model in which cells proliferate very rapidly early in life. To understand how such a saturation effect influences clone size distributions, we introduce an upper limit on birth rate that limits proliferation in the absence of competition. Following *De Boer and Perelson, 1995*, we set the clonal birth rate to $b(t) = b_0/(K + N)$ for some constant $K$, which sets the repertoire size below which competition is negligible. Given this choice the birth rate remains limited to a value $b_0/K$ even in the absence of any competitors. Increasing $K$ leads to deviation in the scaling of the largest clones, but the same scaling remains at intermediate clone sizes (*Figure 2—figure supplement 2*). In the model early clonal growth is exponential until the total repertoire has reached size $N(t) \sim K$, which explains the different distribution of the largest clones. However, the number of clones that are recruited during this phase grows only logarithmically with $K$ due to the exponential increase in total repertoire size.

#### Competition for specific resources

T cells respond to stimuli from peptide-MHC complexes, which could also act as limiting resources. T cells then compete only with those cells specific to the same antigens in contrast to the global competition considered previously. To assess how assumptions about the mechanisms of competition influence, our results we simulated the repertoire formation process using a classical description of competition for antigens (*De Boer and Perelson, 1994*; *De Boer et al., 2001*; *Mayer et al., 2015*). We consider a fixed number of antigens $N_a$ and encode the specificity of the $M$ clones in a matrix $K$ of size $M \times N_a$, where $K_{ij} = 1$ if clone $i$ recognizes the antigen $j$ and $K_{ij} = 0$ otherwise. We draw the entries of $K$ independently with a fixed binding probability $p_b$. We assume that the proliferation rate of a cell of the $i$-th clone is proportional to the amount of antigenic stimulation:

$$b_i = \frac{b_0}{N_a} \sum_{j=1}^{N_a} K_{ij} F_j \tag{50}$$

where the availability of antigen $j$ is given by

$$F_j = \frac{1}{1 + \sum_i K_{ij} C_i}. \tag{51}$$

The normalization of *Equation 50* ensures that total proliferation is comparable to a global resource model with the same parameters independent of $N_a$. For computational tractability, we simulated the clone size dynamics without taking into account demographic stochasticity in proliferation and death of cells. While more specific competition (smaller $p_b$) leads to a deviation in the distribution of the largest clones, we find that clone size distributions are heavy tailed independently of the choice of $p_b$ and all display the same scaling at intermediate clone sizes (*Figure 2—figure supplement 3*).

#### Variations of the recruitment size

The numbers of cells $C_0$ that are recruited might also be variable. In particular, this will be the case when modeling the memory compartment, in which $C_0$ represents the number of cells from a clone recruited into memory following infection. To understand how such variations modify the dynamics of repertoire formation, we derive an analytical prediction in the case where the distribution of recruitment sizes, $P(C_0)$, is lognormal. Given a lognormal distribution with parameters $\mu_0$ and $\sigma_0$ the

mean introduction size is given by $\bar{C}_0 = e^{\mu_0 + \sigma_0^2/2}$. To keep the mean introduction size constant while changing the variability of clone sizes, we use a parametrization in terms of $\bar{C}_0$ and $\sigma_0$ and set $\mu_0 = \log(\bar{C}_0) - \sigma_0^2/2$. To determine the clone size distribution resulting from early repertoire growth, we integrate the continuum theory prediction, $P(C/C_0) \propto (C/C_0)^{-2-\gamma}$ over the distribution of $C_0$:

$$
\begin{aligned}
P(C) \quad &\propto \int_0^C dC_0 (C/\bar{C}_0)^{-2-\gamma} \frac{1}{\sigma_0 C_0/\bar{C}_0 \sqrt{2\pi}} e^{-(\log(C_0/\bar{C}_0) + \sigma_0^2/2)^2/(2\sigma_0^2)} \\
&= (C/\bar{C}_0)^{-2-\gamma} \cdot e^{\frac{1}{2}(\gamma+2)(\gamma+1)\sigma_0^2} \frac{1}{2} \left( \frac{-2\log(C/\bar{C}_0) + (3+2\gamma)\sigma_0^2}{2\sqrt{2}\sigma_0} \right).
\end{aligned}
\tag{52}
$$

The complementary error function $(x)$ saturates for $x \ll -1$. Thus, the distribution follows the same power-law $P(C) \sim C^{-2-\gamma}$ for large clones, $\log C/\bar{C}_0 \gg \sigma_0 \left( \sqrt{2} + \frac{3+2\gamma}{2}\sigma_0 \right)$, while it deviates for smaller clones within the range of recruitment sizes (*Figure 2—figure supplement 1*).

## Appendix 4

### A unified view on mechanisms generating power laws in different growth processes

The origin of power-law scaling during repertoire formation is reminiscent of a class of stochastic processes widely studied in the literature as a mechanism underlying power-law distributions found in diverse contexts (**Yule, 1924**; **Luria and Delbrück, 1943**; **Barabasi and Albert, 1999**), which has been rediscovered multiple times since the pioneering work of Yule on speciation (**Yule, 1924**). Common to these processes is that the distribution of types at a given point is the result of a balance between the growth of existing types and the addition of new types. The different models depending on their context differ in (i) the growth rate $r(t)$ of the number of units of each already existing type and (ii) the rate function $\theta(t)$ at which new types are introduced. They all share the same basic mathematical mechanism that produces a power law distribution of types as we review below. The three maybe most well-known instances of this class of processes are the following:

- The Yule model of speciation (**Yule, 1924**), in which (i) species within a genus speciate at some constant rate, and (ii) new genus is created at a rate proportional to the number of already existing genera.
- The Luria-Delbrück model of population genetics during exponential growth (**Luria and Delbrück, 1943**) in which (i) each cell divides at a constant rate, and (ii) new alleles arise through random mutation at a constant rate per cell division.
- The Barabási-Albert model of network growth (**Barabasi and Albert, 1999**), in which at every time step (ii) a new node is added, and is (i) linked to $m$ already existing nodes with a probability proportional to the number links that a chosen node already has.

Despite differing in their assumptions about the functional form of the growth and innovation rates we show in the following that these different models all share a common mathematical basis. To provide a common terminology, we will use the language of urn models and refer to different types as urns and to the different number of units of each types as balls in each urn. In an attempt to unify the different models, we develop a continuum theory for these growth-innovation processes. To do so, we rescale time to

$$\tau = \int_0^t \theta(t')\mathrm{d}t', \tag{53}$$

such that new urns are added at unit rate, $\theta(\tau) = 1$. The number of balls in each urn then grows according to

$$\frac{\mathrm{d}C_i}{\mathrm{d}\tau} = \frac{\mathrm{d}C_i}{\mathrm{d}t}\frac{\mathrm{d}t}{\mathrm{d}\tau} = \frac{r(t(\tau))}{\theta(t(\tau))}C_i =: \zeta(\tau)C_i. \tag{54}$$

The key to the power-law scaling in all these models is the existence of a regime in which

$$\zeta(\tau) = \frac{1}{\alpha\tau}, \tag{55}$$

that is the growth rate scales inversely with rescaled time with a proportionality factor $1/\alpha$. **Equation 55** has the same form as **Equation 18** that we derived for our model of repertoire formation. Thus following the derivation of **Equation 19** within Materials and methods – Continuum theory of clonal growth we obtain a subexponential growth of balls in already existing urns, which leads to a power law scaling of the distribution of balls per urn,

$$P(C) \propto C^{-\alpha-1}, \tag{56}$$

with an adjustable exponent that depends on $\alpha$.

Before deriving how **Equation 55** arises in specific contexts let us first remark on a general consequence of this form of effective growth law: The total number of balls added to all existing urns per rescaled time unit is constant. To derive this let us assume that each new urn is populated by $C_0$ balls, then the total number of balls $N(t) = \sum_i C_i$ grows according to

$$\frac{dN}{d\tau} = \frac{1}{\alpha\tau}N + C_0, \tag{57}$$

which is solved by

$$N(\tau) = \frac{C_0\alpha}{\alpha - 1}\tau. \tag{58}$$

Multiplying by the growth rate *Equation 55* yields a constant,

$$N(\tau)\zeta(\tau) = \frac{C_0}{\alpha - 1}, \tag{59}$$

thus showing the equivalency between the assumed growth rate dependency on rescaled time and the constancy of how many balls are added per rescaled time unit.

In the Yule process, we have $r(t) = r$ and recruitment is proportional to the number of genera, $\theta(t) = sG(t)$, which grow at a (generally different) rate $s$, $G(t) = G_0 e^{st}$. By integration we obtain $\tau(t) = G_0(e^{st} - 1)$, which leads to $\zeta(\tau) = \frac{r}{s\tau + sG_0}$. Thus $\zeta(\tau) \approx \frac{r}{s\tau}$ when the number of newly created genera exceeds the initial number $\tau \gg G_0$. The exponent of the power-law is determined by the ratio of the growth of genera and species, $\alpha = s/r$.

In the Luria-Delbrück model, we have $r(t) = r$ assuming mutations do not change the growth rate. Recruitment is proportional to the total population size $\theta(t) = \mu r N(t)$, where $\mu$ is the mutation probability per replication and where $N(t) = N_0 e^{rt}$. By integration we obtain $\tau(t) = \mu N_0(e^{rt} - 1)$, which leads to $\zeta(\tau) = \frac{1}{\tau + \mu N_0}$. Thus $\zeta(\tau) = \frac{1}{\tau}$ when $\tau \gg \mu N_0$. In contrast to Yule's model, the power-law exponent is fixed at $\alpha = 1$ under our assumption that mutations are neutral, because the same growth process governs the increase in $\theta(t)$ and in cell numbers.

In the Barabasi-Albert model, the introduction rate $\theta(t) = 1$ is constant, but $r(t)$ decreases with time. The $m$ newly added links attach preferentially to those nodes that already have a large degree. The growth rate $r(t) = m/N(t)$ of a node thus decreases proportionally to the total degree $N = 2mt$ of all present nodes. We have $r(t) = 1/(2t)$, which implies $\zeta(\tau) = \frac{1}{2\tau}$ and $\alpha = 2$.

## Appendix 5

### Relation between clone size and cellular phenotypes

In both cohorts, all T cells from peripheral blood were sequenced irrespective of their phenotypes. Antigenic challenges drive large clonal expansions and we thus expect clones with effector or memory cells to be larger than naive clones all else being equal (*Farber et al., 2014*; *Mayer et al., 2019*). This has generally been confirmed by TCR repertoire sequencing studies (*Oakes et al., 2017*), but there have also been some reports (*Qi et al., 2014*; *Pogorelyy et al., 2017*) of expanded naive clones with similar sizes to the largest memory clones. Given this unclear picture from the literature, we analyzed the relative contribution of naive and memory cells to clones of different sizes.

Overall, we might expect that naive clones dominate the clone size distribution at the smallest sizes. To test this idea, we compared sequencing and flow cytometry data from the Britanova cohort and found that the fraction of naive cells in different individuals explains a remarkably high 88% of variability in the number of clones sequenced only once after subsampling all repertoires to the same size (*Figure 1—figure supplement 2A*). To further determine how cells from clones of different sizes partition phenotypically, we analyzed data from a study in which T cells were sequenced both in unsorted blood as well as after sorting into naive and memory cells (*Chu et al., 2019*). We find that the sizes of large clones follow the same scaling in unsorted blood and in the memory compartment (*Figure 1—figure supplement 2B*) Within the naive compartment most clones are small, in particular when excluding clones from which cells are also found in the memory compartment (*Figure 1—figure supplement 2B*, red line). We note from the plot that all the largest 200 clones in unsorted blood have memory phenotype cells, and less than 1% of the top 1000 clones are not found within the memory compartment. This rules out that the enrichment of zero insertion clones among the most abundant clones found in *Figure 3* is driven by naive clones as has been suggested in a previous study (*Pogorelyy et al., 2017*). The relative frequency of a clone within the memory compartment is larger by a constant fold-factor (*Figure 1—figure supplement 2C*), likely reflecting an increased relative frequency of the large clones when excluding naive cells from the denominator.

Our analysis of the clone size distributions in unsorted blood shows that the largest clones take up an increasing fraction of all sampled reads with age (*Figure 1—figure supplement 3B,E*). Flow cytometry data shows that the overall fraction of memory T cells in blood also increases with age (*Figure 1—figure supplement 2D*; *Pediatric AIDS Clinical Trials Group et al., 2003*; *Britanova et al., 2016*). How much is the former increase explained by the latter? In the following, we show how to answer this question in the absence of direct data on sorted memory T cell repertoires. Let us define $C_i^+$ as the number of memory cells of the $i$-th clone and $C_i^-$ as the number of naive cells. What we are measuring in unsorted blood is the overall fractional size $f_i$ of the clone,

$$f_i = \frac{C_i^+ + C_i^-}{\sum_i C_i^+ + \sum_i C_i^-}. \tag{60}$$

What we are interested in instead is the fraction of the memory compartment taken up by cells belonging to the same clone,

$$f_i^+ = \frac{C_i^+}{\sum_i C_i^+}. \tag{61}$$

We have shown above that empirically for the largest clones $C_i^+ \gg C_i^-$. Thus

$$f_i^+ \approx f_i / r^+, \quad \text{with } r^+ = \frac{\sum_i C_i^+}{\sum_i C_i^+ + \sum_i C_i^-}, \tag{62}$$

where $r^+$ is the fraction of memory cells within the repertoire. Finally, we approximate $r^+$ by its mean value expected at each given age as fitted from the flow cytometry data (*Figure 1—figure supplement 2D*). We find that this normalization mostly collapses the tails of empirical clone size distributions (*Figure 1—figure supplement 3C,F*). This implies that most of the increase in the sizes of the largest clones is explained by the overall expansion of the memory compartment, while the relative clone sizes within the memory compartment are more stable.

