## [Decision Letter]

**Acceptance summary:**

This paper provides a mechanistic model to explain the generation and maintenance of the "power-law" clone size distribution in the memory T cell repertoire. In particular, the authors show that clonal expansions during a perinatal time window are a dominant determinant of T cell clonal abundances- a mechanistic explanation for a long-standing puzzle in quantitative immunology.

**Decision letter after peer review:**

Thank you for submitting your article "Early life imprints the hierarchy of T cell clone sizes" for consideration by *eLife*. Your article has been reviewed by three peer reviewers, and the evaluation has been overseen by a Reviewing Editor and Naama Barkai as the Senior Editor. The following individuals involved in review of your submission have agreed to reveal their identity: Andrew J Yates (Reviewer #1); Herbert Levine (Reviewer #2); Rob J de Boer (Reviewer #3).

The reviewers have discussed the reviews with one another and the Reviewing Editor has drafted this decision to help you prepare a revised submission.

This manuscript provides potential mechanistic explanation for the distribution of memory T cell clone sizes in humans, which follow the same power law in two independent studies. The reviewers all agree that this is an excellent paper, and it presents a thorough explanation of the concepts and the analysis. However, there are still a few comments that we would like to see addressed in this manuscript.

Essential revisions:

1) The basic mechanism described in the manuscript relies on the existence of elevated proliferation early in life, which the authors invoke by postulating that proliferation rates depend inversely on pool size through a competitive mechanism. As far as I am aware there is no strong evidence for such competition, at least under normal (lympho-replete) conditions. As evidence you cite the Douek data showing constant TREC frequencies in blood in children as thymic export wanes, implying reduced proliferation over time (which was analyzed by Bains et al. 2009) – but this was from naive T cells only. Can you find stronger evidence (e.g. Ki67 expression in memory subsets with age) to support this (key) assumption for memory?

In mice, Ki67 across all thymocyte subsets is elevated early in life and settles by about 3mo of age – suggesting that naive clones exported early in life are larger. If such a mechanism operates in humans it could add to the zero-insertion effect that boosts naive T cell clone sizes early in life.

2) On similar lines, you assume that theta (the rate of recruitment of new clones into memory) is independent of age. But thymic involution means the rate of generation of new naive clones wanes with age – might this lead to a reduction in theta? Can you achieve the same results with age-dependent generation of new naive clones and age-dependent recruitment into memory, rather than variation in homeostatic division?

3) The reviewers suggest that the authors provide more details (both immunological and mathematical) in the main text. Make clear whether or not CD4^+^ and CD8^+^ T cells are distinguished by sorting. There is no such thing as "the human T cell repertoire", there are naive and memory, CD4 and CD8 T cell repertoires, and there is a Treg repertoire with a potentially very different clone size distribution. I do agree, however, that your interesting results could be general, i.e., relevant for several of these repertoires, but please make clear for which ones.

Examples from the main text that are confusing on their own include:

i) It is not clear from the main text if are you describing naive or memory T cell repertoires? For an immunologist this makes an enormous difference. For instance, when I read in the Introduction that "At the same time the most abundant clones typically account for more than 1% of all sequencing reads, equivalent to a clone size of ∼10^10^ cells when extrapolating to the full repertoire. It is unknown when these large clonal expansions happen, and more broadly what determines the hierarchy of clone sizes.", I immediately think of the several papers showing that CMV specific memory T cell comprising more than 10% of the repertoire. So, how can I appreciate this "It is unknown when these large clonal expansions happen"? First show that these are not simply CMV specific expansions occurring after CMV infection.

ii) When it is written in the legend of Figure 1 that "Clone sizes were normalized by the […] the memory cell fraction […] in the subset composition of peripheral blood", I read something that cannot be understood without reading the supplementary information. Even after reading the supplementary information I remain confused, because naive and memory repertoires probably have different clone size distributions, so how can one normalize by just knowing the fraction of memory cells?

iii) In the legend of Figure 2 parameter names and values are provided, without giving these parameters interpretation. Please add these interpretations to the main text, because a reader may want to question these values (now it is only explained in the supplementary information). For instance, a cell death rate of 0.2 per year assumes that memory T cells have an expected life span of 5 years. This could be fair given all he uncertainties (e.g., circulating but memory T cells have an expected life span of about half a year, tissue resident memory T cells probably many years, and which subset go memory cells are you sampling from). What is γ? Generating 10^6^ clones per year could be a questionable number (some will say 10^6^ per day, and you confirm this in the supplementary information).

---

## [Author Response]

Essential revisions:1) The basic mechanism described in the manuscript relies on the existence of elevated proliferation early in life, which the authors invoke by postulating that proliferation rates depend inversely on pool size through a competitive mechanism. As far as I am aware there is no strong evidence for such competition, at least under normal (lympho-replete) conditions. As evidence you cite the Douek data showing constant TREC frequencies in blood in children as thymic export wanes, implying reduced proliferation over time (which was analyzed by Bains et al. 2009) – but this was from naive T cells only. Can you find stronger evidence (e.g. Ki67 expression in memory subsets with age) to support this (key) assumption for memory?In mice, Ki67 across all thymocyte subsets is elevated early in life and settles by about 3mo of age – suggesting that naive clones exported early in life are larger. If such a mechanism operates in humans it could add to the zero-insertion effect that boosts naive T cell clone sizes early in life.

We thank the reviewers for raising this important question. The general existence of competition between T cells for homeostatic proliferation triggers has been relatively well accepted within the field for both the naive and memory compartment, see e.g. the reviews by Freitas and Rochas (1) or by Surh and Sprent (2) for a discussion of experimental evidence and possible mechanisms. Here, we thus focus on the more specific question of whether reduced competition early in life leads to elevated proliferation of T cells. While unfortunately we did not find any Ki67 expression data on human memory T cells in infancy in the literature, there are a number of other lines of evidence from both mice and human consistent with the hypothesis of elevated proliferation of T cells early in life. For example, Rufer et al. (3) have shown rapid telomere shortening among memory T cells early in life consistent with large proliferative turnover. Furthermore, early studies of homeostatic proliferation of T cells in neonatal mice have shown that the dividing T cells eventually express memory-like markers (4, 5). Additionally, homeostatic proliferation was shown in these studies to be suppressed more readily by the transfer of memory than naive cells, consistent with competition limiting proliferation specifically within the memory compartment. Following on these early studies there have been a number of reports demonstrating the existence of memory-phenotype antigen-specific cells in unimmmunized or germ-free mice (6, 7), which are thought to result from proliferation of T cells during neonatal lymphopenic conditions stimulated by other antigens. (We have added citations to these studies to our revised manuscript.) Unexposed humans similarly have pre-existing memory-phenotype cells against many viral antigens (8), which might plausibly have the same origin. It seems likely that similar proliferation happens in normal pathogen-replete environments for memory cells created in response to foreign antigens.

Finally, we note that while the specific model we present in the Results section assumes that early life proliferation happens within the memory compartment, the mechanism we present is more general. As we note in the discussion an alternative scenario would be that homeostatic proliferation early in life sets up a broad distribution of naive T cell clone sizes, which is translated upon antigen exposure into a broad distribution of memory T cell clone sizes. Distinguishing these two scenarios is an important direction for future studies.

2) On similar lines, you assume that theta (the rate of recruitment of new clones into memory) is independent of age. But thymic involution means the rate of generation of new naive clones wanes with age – might this lead to a reduction in theta? Can you achieve the same results with age-dependent generation of new naive clones and age-dependent recruitment into memory, rather than variation in homeostatic division?

A temporally changing recruitment rate alone cannot explain the early emergence of power-law scaling of clone sizes, as the distribution of clone sizes is independent of *θ* in a neutral model without homeostatic expansion. We agree that *θ* is likely a declining function of age, both for the naive compartment (because of thymic involution) as well as for the memory compartment (because of a decreasing rate of novel primary infections). Our assumption of a constant *θ* is motivated by a separation of timescales argument: The emergence of scaling during early life only depends on the value of *θ* during roughly the first year of life, and it thus seemed justified not to model the slower dynamics of *θ* due to thymic involution during adulthood.

We have provided an extended justification of this simplifying assumption in the Materials and methods:

"While *θ* decreases during adulthood for both the naive compartment (as thymic production wanes with age) and the memory compartment (as new primary responses are rarer at advanced age), we used a constant *θ* for simplicity. We expect this simplifying assumption not to qualitatively affect the results in Figure 2 due to a separation of timescale: The clone size scaling emerging from the expansionary dynamics only depends on the rate *θ* during infancy and not on slower changes in *θ* happening during adulthood. "

3) The reviewers suggest that the authors provide more details (both immunological and mathematical) in the main text.

We thank the reviewers for their valuable suggestions and have updated the main text to provide more details both on the biology and the mathematics. In addition to the specific suggested changes we have expanded the mathematical derivations underlying the last section of the results, which were previously relegated to the supplementary information. We hope that with these changes our paper strikes a better balance between conciseness and providing necessary context directly in the main text.

Make clear whether or not CD4^+^ and CD8^+^ T cells are distinguished by sorting. There is no such thing as "the human T cell repertoire", there are naive and memory, CD4 and CD8 T cell repertoires, and there is a Treg repertoire with a potentially very different clone size distribution. I do agree, however, that your interesting results could be general, i.e., relevant for several of these repertoires, but please make clear for which ones.

We have updated the caption of Figure 1 to reflect that both studies do not distinguish T cells by sorting (neither by CD4/CD8, naive/memory, or Treg/Tconv).

"Each line shows the size distribution of all T cell clones in an individual in an unsorted blood sample, this is independently of the phenotypes of the cells making up the different clones (see Appendix 5 for analyses of phenotypically resolved data). "

We have also now make more explicit reference to our analyses of phenotypically sorted data in the main text:

"In both studies T cells were sequenced without regard to their phenotypic characteristics. The clone sizes thus represent the full lineage of cells with a common origin irrespective of differentiation status (see Appendix 5 – Relation between clone size and cellular phenotypes and Figure 1—figure supplement 2 for a phenotypically resolved analysis). "

In short, these analyses show that the tail behavior in the unsorted data most closely reflects scaling within the memory compartment. This observation motivates the focus on memory dynamics in the Results section. Nevertheless, we agree that our results might be more general, which is why we present our initial conceptual model without a specific reference to a particular T cell subset. We hope that recent technological advances in coupled single cell phenotyping and T cell sequencing, as well as in bulk repertoire sequencing of sorted T cell populations (9, 10) will help further refine our understanding of how the broad patterns in unsorted blood reflect the composition of various T cell compartments.

Examples from the main text that are confusing on their own include:i) It is not clear from the main text if are you describing naive or memory T cell repertoires? For an immunologist this makes an enormous difference. For instance, when I read in the Introduction that "At the same time the most abundant clones typically account for more than 1% of all sequencing reads, equivalent to a clone size of ∼10^10^ cells when extrapolating to the full repertoire. It is unknown when these large clonal expansions happen, and more broadly what determines the hierarchy of clone sizes.", I immediately think of the several papers showing that CMV specific memory T cell comprising more than 10% of the repertoire. So, how can I appreciate this "It is unknown when these large clonal expansions happen"? First show that these are not simply CMV specific expansions occurring after CMV infection.

We have updated the paragraph in the Introduction to give a more precise motivation of what we aim to explain in our paper:

"In a typical sample of T cells from peripheral blood a large fraction of clones are only seen once within 10^5^−10^7^ sampled sequences, while the most abundant clones account for more than 1% of all sequencing reads. Such power-law scaling of clone sizes has been shown to arise at steady state in models of fluctuating clonal selection as driven by different antigen encounters (Desponds et al., 2016). However, it is unclear whether this mechanism alone is sufficient to explain how clone size scaling is established, and more broadly how variable the clonal hierarchy is over time."

Indeed, what is surprising about the empirical data from a theory perspective is not the existence of some large clones, but rather the reproducible scaling law linking clonal rank and size across individuals with varied history of exposure to different pathogens.

We have also included additional details in the main text to highlight that the cohort studies sequence unsorted T cells from blood samples:

"[…] we reanalyzed data from two large-scale cohort repertoire sequencing studies of human blood samples, which used fundamentally different sequencing pipelines and thus have different sources of noise (Materials and methods). Both studies sequenced the locus coding for the hypervariable TCR CDR3−*β* chain from unsorted T cells from peripheral blood of healthy human volunteers spanning a large range of ages (Figure 1—figure supplement 1)."

Finally, to address the role of CMV specific expansions in setting up the broad hierarchy, we now more explicitly refer in the main text to a supplementary figure analyzing how scaling exponents depend on CMV status.

"While exponents overall decreased slightly with age, the dependence on age accounted for surprisingly little variation in both cohorts (Figure 1D), including when controlling for sex and cytomegalovirus exposure status (Figure 1—figure supplement 5)."

Our analysis shows that while CMV positive individuals do have a more skewed repertoire on average, CMV negative donors also have a broad distributions of clone sizes.

ii) when it is written in the legend of Figure 1 that "Clone sizes were normalized by the […] the memory cell fraction […] in the subset composition of peripheral blood", I read something that cannot be understood without reading the supplementary information. Even after reading the supplementary information I remain confused, because naive and memory repertoires probably have different clone size distributions, so how can one normalize by just knowing the fraction of memory cells?

In short, this normalization is possible because the tail of the distribution in unsorted blood is completely due to clones with a majority of memory phenotype cells. We have provided extended explanation in the main text as detailed below.

We have updated the caption,

"Normalized clone sizes were defined as the number of reads of a given receptor’s sequence divided by the total number of reads within a sample and a factor equal to the average fraction of T cells with memory phenotype at different ages to account for variations in sampling depth and in the subset composition of peripheral blood, respectively (Figure 1—figure supplement 3)."

and the text

"After normalizing clone sizes to account for variations in sampling depth and for the increasing fraction of T cells of memory phenotype with age (Figure 1—figure supplement 3), we found that.…" to more precisely define how this normalization step is implemented. We have also substantially expanded the rationale for this normalization described in Appendix 5.

"Our analysis of the clone size distributions in unsorted blood shows that the largest clones take up an increasing fraction of all sampled reads with age (Figure 1—figure supplement 3B,E). […] This implies that most of the increase in the raw counts of the largest clones is explained by the overall expansion of the memory compartment. "

iii) In the legend of Figure 2 parameter names and values are provided, without giving these parameters interpretation. Please add these interpretations to the main text, because a reader may want to question these values (now it is only explained in the supplementary information). For instance, a cell death rate of 0.2 per year assumes that memory T cells have an expected life span of 5 years. This could be fair given all he uncertainties (e.g., circulating but memory T cells have an expected life span of about half a year, tissue resident memory T cells probably many years, and which subset go memory cells are you sampling from). What is γ? Generating 10^6^ clones per year could be a questionable number (some will say 10^6^ per day, and you confirm this in the supplementary information).

We have restructured our exposition in the main text for greater clarity. We have expanded the description of the model parameters as follows:

"Following previous work (Bains et al., 2009; Lythe et al., 2016) we assume that T cells proliferate at a rate *b* = *b*_0_*/N* inversely proportional to the total number of cells *N* already present in the repertoire. This assumption leads to increased proliferation early in life before the repertoire has reached its homeostatic size, and it is compatible with a simple mechanistic model of T cell competition (Material and methods subsection “Mechanistic motivation for the competition function”). We further assume that cells die at a rate *d*, and that new clones are recruited at rate *θ* with an initial size *C*_0_. For simplicity, we set *C*_0_ equal to one in the following and we assume constant rates *d* and *θ*."

For easier reference we have repeated the parameter names in the caption of Figure 2. We have also included the formula that relates *γ* to the other model parameters in the main text, as well as given an explicit equation for the dependence of the clonal proliferation rate on repertoire size. We note that the parameters are also defined in the model sketch (panel A of the same figure). To highlight the most relevant parameter choices and how they relate to the observed results, we have added the following paragraph to the main text:

"The power-law tail persisted over multiple decades of aging, much beyond the timescale of cellular turnover, 1*/d* = 5 years, assumed in the simulations. A mathematical analysis shows that the relative timescales of clonal and cellular turnover are controlled by the control parameter *γ* = *θC*_0_*/b*_0_, which is the ratio of the contribution of recruitment and proliferation to overall compartment maintenance (Appendix 2). The long timescale of clonal turnover emerges because we have chosen parameters compatible with the biological parameter regime *γ <* 1 (den Braber et al., 2012; Macallan et al., 2017), where most cell death is balanced by proliferation."

We have also extended the parameter discussion in Materials and Methods to highlight that these parameters are not meant to exactly reproduce those for any particular T cell subset, but rather illustrate the dynamical mechanism in a generally relevant parameter regime.

References

1) A. A. Freitas and B. Rocha. Population biology of lymphocytes: the flight for survival. Annual review of immunology, 18(1):83 111, 2000.

2) C. D. Surh and J. Sprent. Homeostasis of Naive and Memory T Cells. Immunity, 29(6):848 862, 2008.

3) N. Rufer, T. H. Brummendorf, S. Kolvraa, C. Bischoff, K. Christensen, L. Wadsworth, M. Schulzer, and P. M. Lansdorp. Telomere Fluorescence Measurements in Granulocytes and T Lymphocyte Subsets Point to a High Turnover of Hematopoietic Stem Cells and Memory T Cells in Early Childhood. Journal of Experimental Medicine, 190(2):157 167, 1999.

4) A. Le Campion, C. Bourgeois, F. Lambolez, B. Martin, S. Léaument, N. Dautigny, C. Tanchot, C. Pénit, and B. Lucas. Naive T cells proliferate strongly in neonatal mice in response to self-peptide/self-MHC complexes. Proceedings of the National Academy of Sciences, 99(7):4538 4543, 2002.

5) B. Min, R. McHugh, G. D. Sempowski, C. Mackall, G. Foucras, and W. E. Paul. Neonates support lymphopenia-induced proliferation. Immunity, 18(1):131 140, 2003.

6) C. Haluszczak, A. D. Akue, S. E. Hamilton, L. D. S. Johnson, L. Pujanauski, L. Teodorovic, S. C. Jameson, and R. M. Kedl. The antigenspecific CD8 + T cell repertoire in unimmunized mice includes memory phenotype cells bearing markers of homeostatic expansion. Journal of Experimental Medicine, 206(2):435 448, 2009.

7) T. Kawabe, D. Jankovic, S. Kawabe, Y. Huang, P.-h. Lee, H. Yamane, J. Zhu, A. Sher, R. N. Germain, and W. E. Paul. Memory-phenotype CD4 + T cells spontaneously generated under steady-state conditions exert innate TH1-lik effector function. Science Immunology, 9304(June), 2017.

8) L. F. Su, B. A. Kidd, A. Han, J. J. Kotzin, and M. M. Davis. Virus-Specific CD4^+^ Memory-Phenotype T Cells Are Abundant in Unexposed Adults. Immunity, 38(2):373 383, 2013.

9) T. Oakes, J. M. Heather, K. Best, R. Byng-Maddick, C. Husovsky, M. Ismail, K. Joshi, G. Maxwell, M. Noursadeghi, N. Riddell, T. Ruehl, C. T. Turner, I. Uddin, and B. Chain. Quantitative characterization of the T cell receptor repertoire of naive and memory subsets using an integrated experimental and computational pipeline which is robust, economical, and versatile. Frontiers in Immunology, 8(OCT):1 17, 2017.

10) C. Soto, R. G. Bombardi, M. Kozhevnikov, M. Gujral, S. Mallal, and J. E. Crowe. High Frequency of Shared Clonotypes in Human T Cell Receptor Repertoires. Cell Reports, 32(2):107882, 2020.